# Backward conditioning reveals flexibility in infralimbic cortex inhibitory memories

**Nura W Lingawi[1], Billy Chieng[1], R Fred Westbrook[1], Nathan Holmes[1], Mark E Bouton[2], Vincent Laurent[1]\***

[1]School of Psychology, UNSW Sydney, Sydney, Australia; [2]Department of Psychological Science, University of Vermont, Burlington, United States

**\*For correspondence:**
v.laurent@unsw.edu.au

## eLife Assessment

This set of experiments provides **important** knowledge for how the infralimbic cortex is recruited to inhibit behavior after extinction training. The evidence supporting the conclusions is **convincing** with multiple sophisticated behavioral designs providing converging lines of evidence, though reviewers note possible alternative interpretations and limitations of small group sizes in some cases. This work will be of interest to those interested in cortical function, learning and memory, aversive behavior, and/or related psychiatric factors.

**Abstract** While the infralimbic cortex (IL) is recognized as critical for behavioral inhibition, the content of the inhibitory memories stored in this region remains elusive. To probe this content, we examined some of the conditions that allow retrieval and facilitation of an inhibitory memory stored in the IL using optogenetic stimulation in female and male rats. We found that IL stimulation did not facilitate an initial fear extinction session. However, prior experience with fear extinction enabled IL stimulation to facilitate subsequent fear extinction. Importantly, the facilitative effects of IL stimulation were not limited to fear extinction experience, as prior exposure to backward fear conditioning also enabled IL stimulation to enhance later fear extinction. The effects were stimulus-specific and did not depend on the motivational context present during the prior experience, as backward appetitive conditioning allowed IL stimulation to facilitate later fear extinction. Additional experiments ruled out stimulus familiarity as an explanation for the facilitative effects of IL stimulation and demonstrated that IL-mediated facilitations occur in procedures other than fear extinction. Together, these findings demonstrate that the IL stores inhibitory memories that are extremely flexible since they can be retrieved and used across many inhibitory procedures and distinct motivational contexts. These features establish the IL as a critical hub for the flexible application of inhibitory knowledge that allows adaptive responses in dynamic environments.

## Introduction

Animals learn about environmental cues that predict the omission of important biological events (*Sosa, 2024*). The neural mechanisms underlying this inhibitory learning have been extensively investigated through studies of fear extinction, a process in which repeated exposure to a fear-inducing stimulus without its associated threat gradually diminishes the stimulus' capacity to elicit fear responses (*Rescorla, 2000*; *Delamater, 2004*). These studies identified the infralimbic cortex (IL) in rodents as a critical neural hub for fear extinction learning (*Bouton et al., 2021*; *Quirk and Mueller, 2008*; *Milad and Quirk, 2012*; *Maren and Quirk, 2004*; *Lingawi et al., 2019*; *Nett and LaLumiere, 2021*; *Marek et al., 2019*). Disrupting IL function across extinction training impairs extinction learning and increases fear responses during later testing (*Do-Monte et al., 2015*; *Lebrón et al., 2004*; *Sierra-Mercado*

*et al., 2011*; *Laurent and Westbrook, 2009*). Moreover, enhanced firing in IL neurons correlates with improved extinction retrieval (*Milad and Quirk, 2002*; *Burgos-Robles et al., 2007*), while suppressing IL activity undermines this retrieval process (*Laurent and Westbrook, 2009*; *Kim et al., 2016*; *Marek et al., 2018*). Conversely, artificially stimulating IL activity enhances extinction memory retrieval (*Do-Monte et al., 2015*; *Kim et al., 2016*; *Milad et al., 2004*; *Vidal-Gonzalez et al., 2006*), providing compelling evidence for the IL's fundamental role in the fear inhibition learned in extinction. However, these findings leave critical questions about the specific nature of the memory being formed in extinction unanswered. Identifying the precise content of inhibitory learning (e.g. what exactly animals learn when they undergo fear extinction) remains essential for understanding both the mechanisms of inhibition and the broader adaptive functions of the IL in behavioral flexibility.

To study the content of inhibitory learning, we examined some of the conditions that allow retrieval of an inhibitory memory stored in the IL. We reasoned that successful retrieval could be revealed by demonstrating that IL stimulation during memory recall durably enhances the retrieved memory (*Lingawi et al., 2018*; *Lingawi et al., 2017*). We found that pharmacological stimulation of the IL during an initial fear extinction session suppressed fear responses during that session but did not enhance fear inhibition during later testing. However, the same stimulation applied during a *second* extinction session significantly enhanced fear inhibition at test. These results demonstrated that prior fear extinction experience enabled IL stimulation to facilitate fear inhibition during subsequent extinction learning. The facilitation effect proved to be stimulus-specific, occurring only when the same fear-inducing stimulus was used across both extinction sessions. Additional experiments revealed that prior fear extinction was not the only experience that enabled IL stimulation to facilitate fear extinction. Similar facilitative effects emerged when animals were pre-exposed to the stimulus in a latent inhibition procedure or when the stimulus had previously predicted an appetitive outcome, such as food, followed by extinction of that prediction. These findings led us to propose that inhibitory procedures generate stimulus-specific inhibitory memories within the IL that lack motivational information about the omitted event. During subsequent fear extinction, IL stimulation can retrieve and strengthen these pre-existing inhibitory memories, thereby facilitating fear inhibition.

Our proposal aligns with research demonstrating IL involvement across diverse inhibitory learning procedures beyond those discussed above (*Rhodes and Killcross, 2007*; *Fam et al., 2023*; *Ng et al., 2024*; *Krueger et al., 2024*; *Ng and Sangha, 2023*; *Broomer and Bouton, 2024*). The hypothesis that IL memories lack motivational specificity is consistent with evidence showing IL engagement in procedures involving non-threatening outcomes (*Peters et al., 2009*; *Nett and LaLumiere, 2021*; *Laurent et al., 2016*; *Meyer and Bucci, 2014*; *Brown et al., 2023*; *Broomer and Bouton, 2024*), suggesting a general role in inhibitory processing rather than fear-specific mechanisms. Nevertheless, the findings described have at least three limitations that warrant consideration. First, all the first-phase inhibitory procedures involved presenting the target stimulus in isolation, raising questions about whether IL recruitment occurs under different stimulus presentation conditions. Some existing literature suggests this may indeed be a critical factor in IL engagement (*Lay et al., 2020*). Second, prior experience with fear extinction, latent inhibition, or appetitive extinction could have simply increased familiarity with the target stimulus, and this familiarity might have enabled IL stimulation to be effective during subsequent fear extinction. Finally, the facilitative effects of IL stimulation were demonstrated exclusively during fear extinction sessions, limiting the generalizability of the findings to other inhibitory learning procedures.

The present series of experiments addressed these limitations to enhance understanding of inhibitory learning mechanisms and IL function. To determine whether IL engagement extends beyond procedures employing exposures to the target stimulus in isolation, the experiments utilized backward conditioning, during which a target stimulus is reliably presented a few seconds after encountering a biologically significant event. This procedure has been shown to imbue the backward conditioned stimulus with inhibitory properties (*Laurent et al., 2022*; *Laurent et al., 2018*; *Laurent et al., 2015*; *Laurent and Balleine, 2015*; *Delamater et al., 2003*; *Seitz et al., 2022*; *Sosa, 2024*; *Chang et al., 2003*; *Cole and Miller, 1999*; *Heth, 1976*). Thus, the first key experiment in the series examined whether prior experience with backward fear conditioning enables IL stimulation to facilitate subsequent fear extinction. After showing that this is, indeed, the case, subsequent experiments then examined whether the facilitative effects of IL stimulation can be observed following backward *appetitive* conditioning, is due to familiarity of the stimulus that would eventually undergo extinction and

can be observed across different circumstances of testing. Since the current research was interested in uncovering the general role played by the IL in inhibitory learning, it used an optogenetic approach that stimulated all local neurons, regardless of their molecular identity. This broad approach more closely mimicked the pharmacological stimulation used in our previous studies (*Lingawi et al., 2018*; *Lingawi et al., 2017*), while ensuring that the stimulation was restricted anatomically to the IL and temporally aligned with presentations of the target stimulus. The first two experiments, therefore, involved ex-vivo electrophysiological assessments of the capacity to stimulate IL neurons, followed by a replication of the core finding that prior fear extinction experience enables IL stimulation to enhance subsequent fear extinction learning.

## Results

### Ex-vivo cell recordings demonstrate successful optical IL stimulation

We first used ex-vivo electrophysiological recordings to assess our capacity to optically stimulate IL neurons. Wild-type Long-Evans female and male rats were bilaterally infused in the IL with an excitatory channelrhodopsin (ChR2) virus (*Figure 1A*), and recordings were completed in 10 cells transduced with the virus. Overall, LED light illumination (465 nm, 5 ms pulse, 20 Hz) enhanced action potential firing compared to the period preceding illumination (baseline; *Figure 1B*; $F_{(1,9)}$ = 13.69, $p<0.01$; $\eta^2=0.60$). Based on the physiological and morphological characteristics of these cells, we identified one as a non-pyramidal cell, which was excited by LED light illumination (from 0 at baseline to 13.6 Hz under LED). The remaining 9 cells were pyramidal, and one did not respond to LED light illumination, but we cannot exclude the possibility that this was due to a lack of ChR2 expression in the somatic compartment. Another pyramidal cell showed a slight reduction in activity following LED light illumination (from 6 Hz at baseline to 3.6 Hz under LED), but the remaining 7 cells displayed clear excitation upon LED light illumination. Thus, we conclude that ChR2-mediated excitation was substantial at the time of LED light delivery, effectively validating our ability to use ChR2 to stimulate IL neuronal activity.

### Prior experience with fear extinction enables IL stimulation to facilitate subsequent fear extinction

Experiment 1 aimed to confirm that IL stimulation facilitates fear extinction in animals with previous extinction experience (*Lingawi et al., 2017*). Rats received bilateral IL infusions of the ChR2 virus and were implanted with fiber-optic cannulae positioned above the IL (*Figure 2—figure supplement 1A*). All animals then underwent the behavioral procedures described in *Figure 2A*. Following initial forward fear conditioning to a tone (i.e. the tone ended with shock delivery), rats were allocated to two conditions: half (ReExt-OFF and ReExt-ON) received an initial fear extinction session to the tone, while in the other condition (Ext-OFF and Ext-ON) rats were handled only. All rats subsequently underwent a second fear conditioning session followed by a second fear extinction session. In line with our previous work (*Lingawi et al., 2018*; *Lingawi et al., 2017*), the second extinction session in this experiment and the subsequent ones was brief, involving eight tone-alone presentations (six presentations in remaining experiments). The low number of stimulus presentations aimed to generate minimal extinction learning to maximize detection of any facilitatory effects of IL stimulation. During the brief extinction session, LEDs were activated in half the animals from each behavioral condition (Ext-ON and ReExt-ON) but remained inactive in the other half (Ext-OFF and ReExt-OFF). Finally, all rats were tested for freezing to the tone.

Freezing data are presented in *Figure 2B*. Initial forward fear conditioning proceeded as expected, with freezing to the tone gradually increasing across trials (Trial: $F_{(1,34)}$ = 195.08; $p<0.001$; $\eta^2=0.85$). A protocol effect emerged, with rats designated for two extinction sessions (ReExt-OFF and ReExt-ON) showing greater freezing than those assigned to a single session (Ext-OFF and Ext-ON; Protocol: $F_{(1,34)}$ = 4.56; $p<0.05$; $\eta^2=0.12$). Nevertheless, all rats increased their freezing to the tone as training progressed, regardless of protocol assignment (Protocol × Trial: $p=0.09$). No other significant differences were observed (lowest $p=0.45$), including during the baseline period (i.e. prior to the first tone presentation; lowest $p=0.13$).

The first extinction session proceeded smoothly, with tone-elicited freezing gradually decreasing across trial blocks (Block: $F_{(1,17)}$ = 56.225; $p<0.05$; $\eta^2=0.77$), regardless of subsequent LED condition

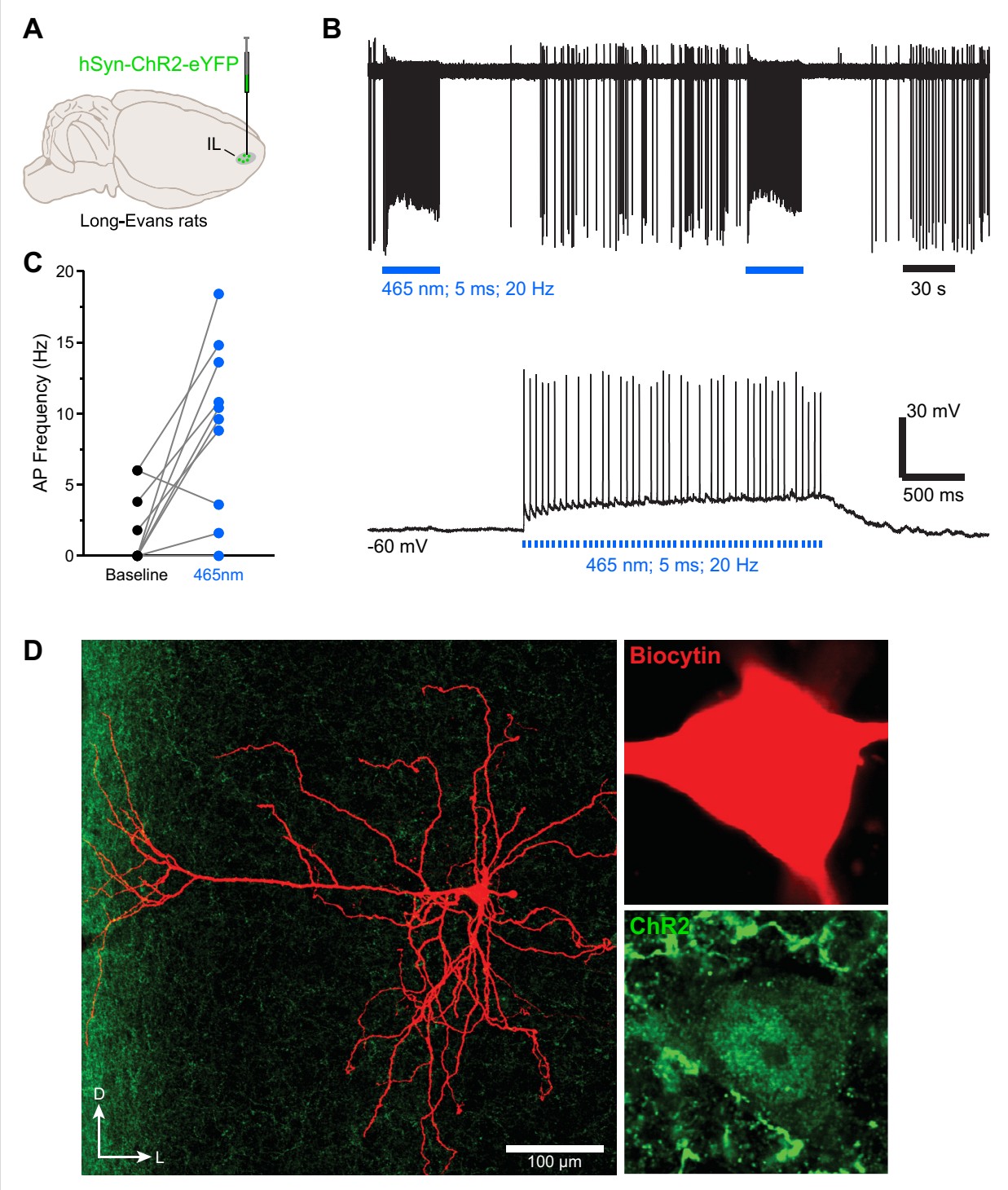

**Figure 1.** Ex-vivo cell recordings demonstrate successful optical infralimbic cortex (IL) stimulation. (**A**) Wild-type Long-Evans rats (n=2 females and 2 males) were bilaterally infused in the IL with a channelrhodopsin (ChR2) virus. (**B**) The top representative raw trace is a cell-attached recording from a transfected IL neuron with LED train stimulations. In the same neuron under whole-cell current clamp, LED pulses excited the IL neuron as evidenced by enhanced action potentials (AP) from resting state baseline (bottom raw trace). (**C**) AP frequency was enhanced during LED illumination compared to their baseline state (n=10 cells). (**D**) Micrographs showing ChR2-transfected IL neurons that was labeled with biocytin.

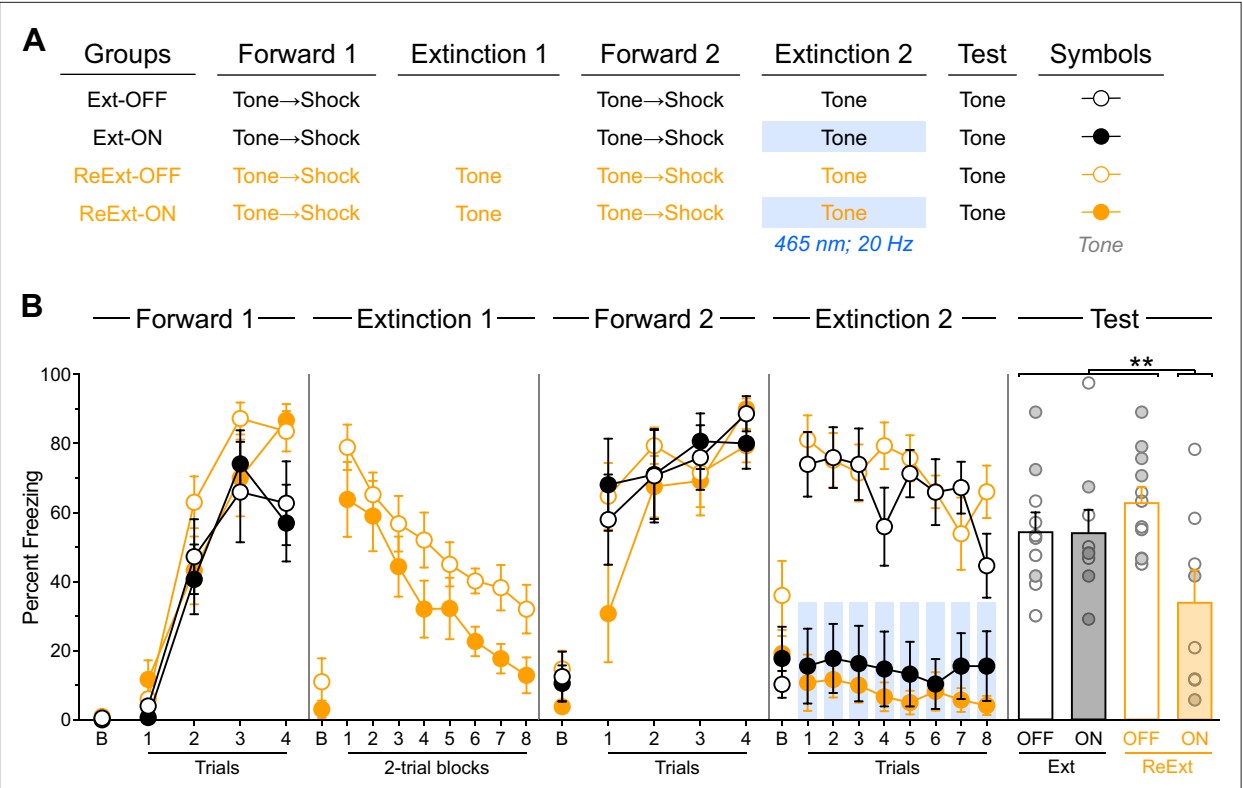

**Figure 2.** Prior experience with fear extinction enables infralimbic cortex (IL) stimulation to facilitate subsequent fear extinction. (**A**) Schematic representation of the behavioral design used in Experiment 1. The LEDs were activated in the Ext-ON and ReExt-ON groups during the second fear extinction session. Ext-OFF: n=3 females and 7 males; Ext-ON: n=4 females and 5 males; ReExt-OFF: n=5 females and 6 males; ReExt-ON: n=2 females and 6 males. (**B**) Mean percent freezing during all experimental stages. Data are shown as mean ± SEM. Test data include individual data points for female (filled circle) and male (open circle) rats. Asterisks denote significant effect (**$p<0.01$).

The online version of this article includes the following figure supplement(s) for figure 2:

**Figure supplement 1.** Histological and behavioral data related to *Figure 2*.

assignment (Block × LED: $p=0.49$). No other significant differences were detected (LED: $p=0.07$), including during the baseline period (LED: $p=0.36$).

The second forward conditioning session was successful, with freezing to the tone increasing across trials (Trial: $F_{(1,34)} = 11.41$; $p<0.01$; $\eta^2=0.25$), regardless of protocol or LED assignment (lowest $p=0.09$). No other significant differences were detected (lowest $p=0.47$), including during the baseline period (lowest $p=0.26$). It is worth noting that these analyses indicate that the effect of protocol observed during initial forward fear conditioning or the trend for lower freezing in the ReExt-ON condition during the first extinction session were short-lived.

The brief extinction session produced a gradual decline in tone-elicited freezing across trials (Trial: $F_{(1,34)} = 12.32$; $p<0.001$; $\eta^2=0.25$), regardless of protocol assignment (Protocol × Trial: $p=0.88$). IL stimulation severely reduced freezing (LED: $F_{(1,34)} = 67.82$; $p<0.001$; $\eta^2=0.67$), regardless of protocol assignment (LED × Protocol: $p=0.39$). Consequently, the gradual decline in freezing across trial blocks was more pronounced in rats that did not receive IL stimulation (LED × Trial: $F_{(1,34)} = 4.81$; $p<0.05$; $\eta^2=0.12$). No other significant differences were observed (lowest $p=0.66$), including during the baseline period (lowest $p=0.09$).

The test data confirmed that IL stimulation during the brief extinction session reduced freezing only in rats that had previously undergone an extinction session (*Figure 2B*, *Figure 2—figure supplement 1*). Rats without prior extinction experience displayed substantial and equivalent freezing levels, regardless of IL stimulation (Ext-OFF vs. Ext-ON: $p=0.96$). These freezing levels were comparable to those observed in rats that had received initial extinction but no IL stimulation during the brief session (Ext-OFF +Ext ON vs. ReExt-OFF: $p=0.25$). Critically, all three of these groups exhibited significantly greater freezing than rats that had received both initial extinction and IL stimulation during

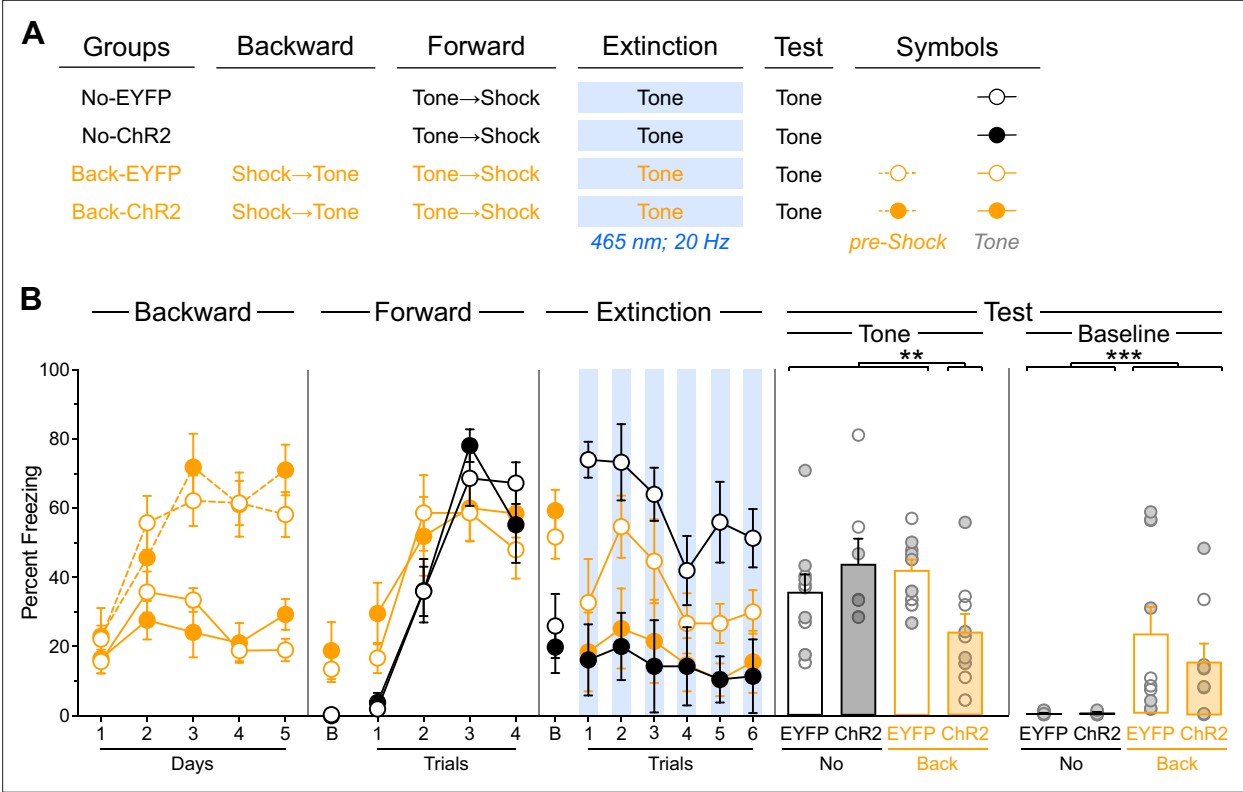

**Figure 3.** Prior experience with backward fear conditioning enables infralimbic cortex (IL) stimulation to facilitate subsequent fear extinction. (**A**) Schematic representation of the behavioral design used in Experiment 2. The LEDs were activated in all groups during the fear extinction session. No-EYFP: n=5 females and 5 males; No-ChR2: n=3 females and 4 males; Back-EYFP: n=5 females and 5 males; Back-ChR2: n=4 females and 5 males. (**B**) Mean percent freezing during all experimental stages. Baseline freezing (before first tone presentation) at test is provided due to significant differences. Data are shown as mean ± SEM. Test data include individual data points for female (filled circle) and male (open circle) rats. Asterisks denote significant effect (**$p<0.01$; ***$p<0.001$).

The online version of this article includes the following figure supplement(s) for figure 3:

**Figure supplement 1.** Histological and behavioral data related to *Figure 3*.

the subsequent brief extinction session (Ext-OFF +Ext ON+ReExt OFF vs. ReExt-ON: $F_{(1,34)} = 9.20$; $p<0.01$; $\eta^2=0.21$). No significant differences emerged during the baseline period (lowest $p=0.21$).

These findings confirm that prior fear extinction experience enables IL stimulation to enhance fear extinction (*Lingawi et al., 2017*). This pattern aligns with the proposal that the IL encodes inhibitory memories formed during initial extinction training, and that IL stimulation during later sessions promotes the retrieval and strengthening of these original inhibitory memories.

## Prior experience with backward fear conditioning enables IL stimulation to facilitate subsequent fear extinction

Experiment 2 tested whether prior experience with backward fear conditioning enables IL stimulation to facilitate fear extinction. Since Experiment 1 showed no effect of LED activation alone, we eliminated this factor and instead included a null virus control. Rats received bilateral IL infusions of a null (EYFP) or ChR2 virus and were implanted with fiber-optic cannulae positioned above the IL (*Figure 3—figure supplement 1A*). As outlined in *Figure 3A*, half the rats in each viral condition (Back-EYFP and Back-ChR2) underwent backward fear conditioning to a tone (i.e. the tone was presented 5 s after shock delivery), while the remaining rats (No-EYFP and No-ChR2) received handling only. Previous research indicates that inhibition in backward conditioning is more likely to develop with prolonged training and with the insertion of a time delay between delivery of the motivational event (e.g. the footshock) and presentation of the stimulus (*Laurent et al., 2022*; *Laurent et al., 2018*; *Laurent et al., 2015*; *Laurent and Balleine, 2015*; *Delamater et al., 2003*;

*Seitz et al., 2022*; *Sosa, 2024*; *Chang et al., 2003*; *Cole and Miller, 1999*; *Heth, 1976*). Thus, backward conditioning in this experiment and the next experiments was administered across five consecutive days and using a 5 s delay between shock presentation and tone onset. After this conditioning, all rats underwent forward tone fear conditioning followed by a brief fear extinction session with LED activation during all tone presentations. Finally, all rats were tested for freezing to the tone.

Freezing data are presented in *Figure 3B*. Backward fear conditioning proceeded as expected (*Laurent et al., 2022*), with overall freezing increasing across days (Day: $F_{(1,17)}$ = 27.54; $p<0.001$; $\eta$²=0.62). Rats showed greater freezing during the pre-shock periods compared to tone presentations (Period: $F_{(1,17)}$ = 76.09; $p<0.001$; $\eta$²=0.82) with this discrimination becoming larger as training progressed (Period × Day: $F_{(1,17)}$ = 174.68; $p<0.001$; $\eta$²=0.91). The virus assignment had no significant influence on these effects (lowest $p$=0.13).

Forward fear conditioning was successful, with freezing to the tone gradually increasing across trials (Trial: $F_{(1,32)}$ = 99.44; $p<0.001$; $\eta$²=0.76). However, this increase was less pronounced in rats that had received backward conditioning (Trial ×Protocol: $F_{(1,32)}$ = 14.72; $p<0.001$; $\eta$²=0.32), as these animals displayed elevated freezing during the first tone trial. This pattern suggests that backward conditioning generated modest excitatory associations between the tone and shock (*Cole and Miller, 1999*). Evidence for backward conditioning effects was further supported by the baseline period analysis, where rats with backward conditioning experience showed significantly greater context freezing (Protocol: $F_{(1,32)}$ = 11.66; $p<0.01$; $\eta$²=0.27). This indicates that backward conditioning established excitatory associations between the context (i.e. conditioning chamber) and shock (*Chang et al., 2003*). No other significant differences were observed during this stage (lowest $p$=0.45).

The brief extinction session produced a gradual decline in freezing to the tone across trials (Trial: $F_{(1,32)}$ = 17.16; $p<0.001$; $\eta$²=0.55), regardless of protocol assignment (Protocol×Trial: $p$=0.63). Consistent with Experiment 1, IL stimulation reduced tone freezing (Virus: $F_{(1,32)}$ = 14.02; $p<0.001$; $\eta$²=0.30), regardless of protocol assignment (Virus×Protocol: $p$=0.11). However, unlike the previous experiment, the gradual decline in freezing across tone trials was equivalent between IL-stimulated and non-stimulated rats (Virus×Trial: $p$=0.07). Visual inspection of the data suggests this lack of difference resulted from reduced freezing in non-IL-stimulated rats that had received backward fear conditioning. This reduction was not unexpected, given that inhibitory stimuli acquire excitatory properties more slowly than neutral stimuli (*Rescorla, 1969*), and we anticipated that backward fear conditioning would confer inhibitory properties to the tone. A planned a priori analysis confirmed that non-IL-stimulated rats showed reduced freezing when they had received backward conditioning (Back-EYFP vs. No-EYFP: $F_{(1,32)}$ = 13.58; $p<0.001$; $\eta$²=0.30). Consistent with the previous stage, baseline period analysis revealed greater context freezing in backward-conditioned rats (Protocol: $F_{(1,32)}$ = 19.82; $p<0.001$; $\eta$²=0.38), confirming substantial excitatory associations between the context and shock. No other significant differences were observed during this stage (lowest $p$=0.11).

The test data revealed that IL stimulation during the brief fear extinction session reduced freezing only in rats that had undergone backward conditioning (*Figure 3B*, *Figure 3—figure supplement 1B*). Rats without prior backward experience displayed substantial and equivalent freezing levels regardless of IL stimulation (No-EYFP vs. No-ChR2: $p$=0.28). These freezing levels were comparable to those observed in rats that had received backward conditioning but no IL stimulation during the brief extinction session (No-EYFP +No-ChR2 vs. Back-EYFP: $p$=0.70). Critically, all three of these groups exhibited significantly greater freezing than rats that had received both backward conditioning and IL stimulation during the brief extinction session (No-EYFP +No-ChR2+Back EYFP vs. Back-ChR2: $F_{(1,32)}$ = 8.15; $p<0.01$; $\eta$²=0.21). The baseline period analysis revealed greater context freezing in rats that received backward conditioning (Protocol: $F_{(1,32)}$ = 13.49; $p<0.001$; $\eta$²=0.30) irrespective of virus assignment (Protocol×Virus: $p$=0.46). The virus assignment had no effect during the baseline period ($p$=0.46).

These findings indicate that backward fear conditioning enables IL stimulation to facilitate subsequent fear extinction. Thus, presenting the target stimulus in isolation does not appear to be the critical factor controlling IL engagement in memory encoding. Rather, these findings are consistent with the hypothesis that backward fear conditioning establishes inhibitory memories in the IL that can be retrieved and strengthened during subsequent fear extinction sessions through local optogenetic stimulation.

## Backward fear conditioning produces an IL memory that is specific to the backward trained stimulus and is inhibitory

Experiment 3 addressed two key questions. First, it tested whether the facilitative effect of IL stimulation following backward fear conditioning operates through modulation of baseline contextual freezing, since differences in such freezing were present in the previous experiment. Second, it examined the stimulus specificity of the IL-mediated facilitation by determining whether the effect is restricted to the stimulus that underwent backward fear conditioning. All rats received bilateral IL infusions of either EYFP or ChR2 virus and were implanted with fiber-optic cannulae targeting the IL (*Figure 4—figure supplement 1A*). Following the procedures shown in *Figure 4A*, rats in each viral group were randomly assigned to one of two backward conditioning procedures: half received backward conditioning with a light (Diff-EYFP and Diff-ChR2 groups), while the other half received identical conditioning using a tone (Same-EYFP and Same-ChR2 groups). Subsequently, all rats underwent standard forward tone fear conditioning, followed by a brief extinction session during which LED stimulation was delivered concurrent with each tone presentation. Finally, all rats were tested for freezing to the tone.

Freezing data are presented in *Figure 4B*. Backward fear conditioning proceeded as expected, with overall freezing increasing across days (Day: $F_{(1,22)}$ = 35.02; $p<0.001$; $\eta^2$=0.61). Rats showed greater freezing during the pre-shock periods compared to stimulus presentations (Period: $F_{(1,22)}$ = 139.81; $p<0.001$; $\eta^2$=0.86) with this discrimination becoming larger as training progressed (Period×Day: $F_{(1,22)}$ = 98.77; $p<0.001$; $\eta^2$=0.82). An unexpected Protocol×Virus interaction emerged ($F_{(1,22)}$ = 4.43; $p<0.05$; $\eta^2$=0.17). In the rats trained with the light (Diff-EYFP and Diff-ChR2 groups), overall freezing was similar across virus condition (Virus: $p=0.50$). It increased across days (Day: $F_{(1,11)}$ = 22.54; $p<0.001$; $\eta^2$=0.67), it was greater in the pre-shock periods compared to stimulus presentation (Period: $F_{(1,11)}$ = 80.53; $p<0.001$; $\eta^2$=0.88), and the discrimination between the two periods became larger as training progressed (Day×Period: $F_{(1,11)}$ = 59.46; $p<0.001$; $\eta^2$=0.84). None of these effects were influenced by viral assignment (Virus×Day, Virus×Period, Virus×Day×Period: lowest $p=0.28$). In the rats trained with the tone (Same-EYFP and Same-ChR2 groups), overall freezing was greater in those that had been infused with the ChR2 virus (Virus: $F_{(1,11)}$ = 9.49; $p<0.05$; $\eta^2$=0.46). Freezing increased across days (Day: $F_{(1,11)}$ = 13.50; $p<0.001$; $\eta^2$=0.55), it was higher in the pre-shock periods compared to stimulus presentation (Period: $F_{(1,11)}$ = 66.33; $p<0.001$; $\eta^2$=0.86) and the discrimination between the two periods grew as training progressed (Day×Period: $F_{(1,11)}$ = 41.02; $p<0.001$; $\eta^2$=0.79). None of these factors were influenced by virus assignment (Virus×Day, Virus× Period, Virus×Day×Period: lowest $p=0.10$). No other significant differences were observed during this stage (lowest $p=0.06$).

Forward fear conditioning was successful, with freezing to the tone gradually increasing across trials (Trial: $F_{(1,22)}$ = 58.104; $p<0.001$; $\eta^2$=0.73). A Protocol x Virus interaction was again detected ($F_{(1,22)}$ = 9.28; $p<0.01$; $\eta^2$=0.30). In the rats trained with the light (Diff-EYFP and Diff-ChR2 groups), overall freezing was similar across virus condition (Virus: $p=0.22$) and increased across trials (Trial: $F_{(1,11)}$ = 64.18; $p<0.001$; $\eta^2$=0.85), regardless of viral assignment (Virus×Trial: $p=0.52$). In the rats trained with the tone (Same-EYFP and Same-ChR2 groups), overall freezing was greater in rats that had been infused with ChR2 (Virus: $F_{(1,11)}$ = 11.63; $p<0.01$; $\eta^2$=0.51). Nevertheless, freezing increased across trials (Trial: $F_{(1,11)}$ = 12.45; $p<0.01$; $\eta^2$=0.53), regardless of viral assignment (Virus×Trial: $p=0.97$). No other significant differences were observed during this stage (lowest $p=0.06$). During the baseline period, rats that had previously received backward conditioning with the tone displayed greater freezing (Protocol: $F_{(1,22)}$ = 4.84; $p<0.05$; $\eta^2$=0.18). No other significant differences were observed during this period (lowest $p=0.16$). Although some stimulus-related differences appeared, their interpretation remains unclear given their presence in the earlier experimental stage and the expectation that baseline contextual freezing should be independent of stimulus identity.

The brief extinction session produced a gradual decline in tone-elicited freezing across trials (Trial: $F_{(1,22)}$ = 6.33; $p<0.05$; $\eta^2$=0.22), regardless of protocol and viral assignment (Protocol×Trial: $p=0.06$; Virus×Trial: $p=0.23$), and the three factors did not interact (Protocol×Virus× Trial: $p=0.37$). Consistent with previous experiments, IL stimulation reduced tone freezing (Virus: $F_{(1,22)}$ = 38.55; $p<0.001$; $\eta^2$=0.64), regardless of protocol assignment (Virus ×Protocol: $p=0.07$). In contrast to the previous experiment, no significant differences were detected during the baseline period (lowest $p=0.13$).

The test results revealed that backward fear conditioning enables IL stimulation to facilitate subsequent fear extinction only when the same stimulus is employed throughout the procedures

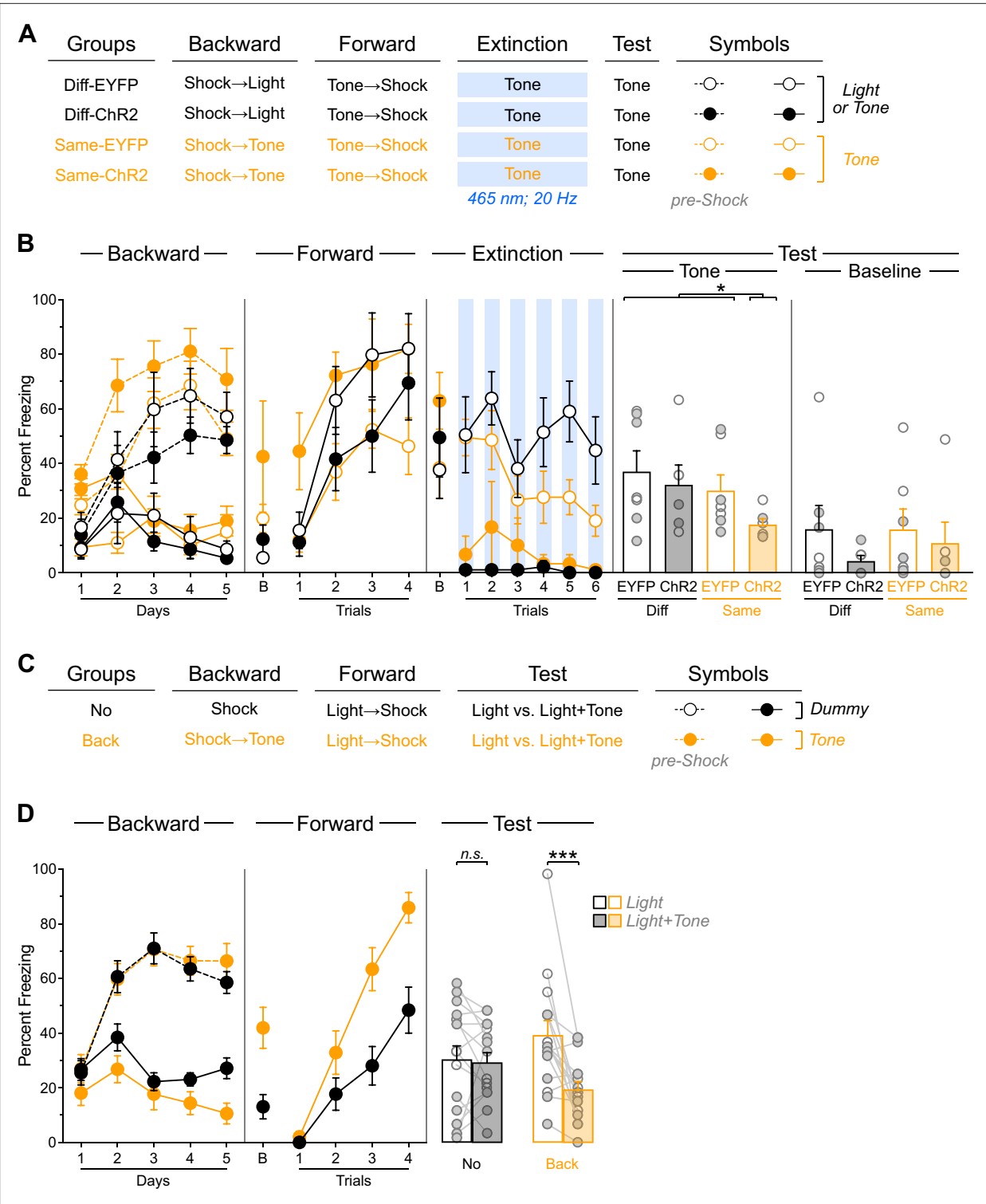

**Figure 4.** Backward fear conditioning produces an IL memory that is specific to the backward trained stimulus and is inhibitory. (**A**) Schematic representation of the behavioral design used in Experiment 3. The LEDs were activated in all groups during the fear extinction session. Diff-EYFP: n=4 females and 3 males; Diff-ChR2: n=2 females and 4 males; Same-EYFP: n=4 females and 3 males; Same-ChR2: n=2 females and 4 males. (**B**) Mean percent freezing during all experimental stages. Baseline freezing (before first tone presentation) at test is provided to compare with Experiment 3. (**C**) Schematic representation of the behavioral design used in Experiment 4. No: n=12 females and 4 males; Back: n=8 females and 8 males. (**D**) Mean percent freezing during all experimental stages. Data are shown as mean ± SEM. Test data include individual data points for female (filled circle) and male (open circle) rats. Asterisks denote significant effect (*p<0.05; ***p<0.001; n.s., nonsignificant).

*Figure 4 continued on next page*

*Figure 4 continued*

The online version of this article includes the following figure supplement(s) for figure 4:

**Figure supplement 1.** Histological and behavioral data related to *Figure 4*.

(*Figure 4B*, *Figure 4—figure supplement 1B*). Rats with prior backward experience to the light displayed substantial and equivalent freezing levels to the tone, regardless of IL stimulation (Diff-EYFP vs. Diff-ChR2: $p=0.28$). These freezing levels were comparable to those observed in rats that had received backward conditioning to the tone but no IL stimulation during the brief extinction session (Diff-EYFP +Diff-ChR2 vs. Same-EYFP: $p=0.54$). Critically, all three of these groups exhibited significantly greater freezing than rats that had received both backward conditioning to the tone and IL stimulation during the brief extinction session (Diff-EYFP +Diff-ChR2+Same EYFP vs. Same-ChR2: $F_{(1,22)} = 4.41$; $p<0.05$; $\eta^2=0.17$). Unlike in the previous experiment, no significant differences were observed during the baseline period (lowest $p=0.26$). It is worth noting, however, that some unexpected interactions emerged across the first two training stages, which were driven by greater freezing in rats that had received backward conditioning to the tone and ChR2 infusion in the IL. Although the source of this greater freezing remains unclear, it seems unlikely to have influenced test performance, since the group concerned expressed lower freezing during that test.

These findings confirm that backward fear conditioning enables IL stimulation to facilitate subsequent fear extinction. Crucially, they also demonstrate that this facilitative effect operates independently of baseline contextual freezing modulation; rather, it is specific to the stimulus that underwent backward conditioning. Thus, backward fear conditioning generates stimulus-specific memories within the IL.

The interpretation of the previous results depends on the hypothesis that backward fear conditioning establishes inhibitory memories that are subsequently retrieved during fear extinction. The ability of backward fear conditioning to confer inhibitory properties to the trained stimulus is supported by results from Experiment 2, where the backward-trained tone showed delayed acquisition of excitatory properties during forward fear conditioning, as evidenced by reduced freezing during brief extinction. This retardation effect was replicated in Experiment 3: the backward fear-conditioned tone elicited less freezing during the brief fear extinction session compared to the neutral tone (Same-EYFP vs. Diff-EYFP: $F_{(1,22)} = 4.65$; $p<0.05$; $\eta^2=0.17$). However, demonstrating inhibition requires additional evidence beyond retardation (*Rescorla, 1969*), and Experiment 4 aimed to provide this by examining whether our backward protocol enables the stimulus to produce subtractive summation—the capacity of an inhibitory stimulus to reduce conditioned responding elicited by an excitatory stimulus. As illustrated in *Figure 4C*, rats were randomly assigned to one of two procedures: rats in one procedure received backward fear conditioning to a tone (Back), while those in the other procedure received context shock exposure without the tone (No). Subsequently, all rats underwent forward fear conditioning to a light, followed by testing the next day. The test compared freezing responses to the light alone versus combined light and tone presentations.

Freezing data are presented in *Figure 4D*. During the initial experimental phase, we monitored freezing during pre-shock periods for both groups, during tone presentations in the backward conditioning group, and during equivalent dummy periods in the control group (No). Overall freezing increased progressively across training days (Day: $F_{(1,30)} = 14.10$; $p<0.001$; $\eta^2=0.32$), regardless of protocol assignment (Protocol×Day: $p=0.192$). Rats showed greater freezing during pre-shock periods compared to tone presentations or dummy periods (Period: $F_{(1,30)} = 197.471$; $p<0.001$; $\eta^2=0.87$), but this difference was more pronounced in rats that had received backward training (Protocol×Period: $F_{(1,30)} = 6.14$; $p<0.05$; $\eta^2=0.17$). The discrimination between periods became larger as training progressed (Period×Day: $F_{(1,30)} = 82.20$; $p<0.001$; $\eta^2=0.73$), regardless of protocol (Protocol× Period×Day: $p=0.19$). Overall freezing levels were comparable across groups (Protocol: $p=0.40$).

Forward fear conditioning to the light proceeded successfully, with freezing increasing gradually across trials (Trial: $F_{(1,30)} = 177.68$; $p<0.001$; $\eta^2=0.86$). The increase was more pronounced in rats with prior backward fear conditioning (Protocol×Trial: $F_{(1,30)} = 14.85$; $p<0.001$; $\eta^2=0.83$), suggesting that excitatory associations between the tone and shock from backward conditioning somehow enhanced fear to the light. Consequently, overall freezing was elevated in rats that had received backward fear conditioning (Protocol: $F_{(1,30)} = 11.71$; $p<0.001$; $\eta^2=0.28$). During the baseline period, a similar pattern emerged, with backward-trained rats displaying greater contextual freezing (Protocol: $F_{(1,30)} = 10.80$; $p<0.001$; $\eta^2=0.26$).

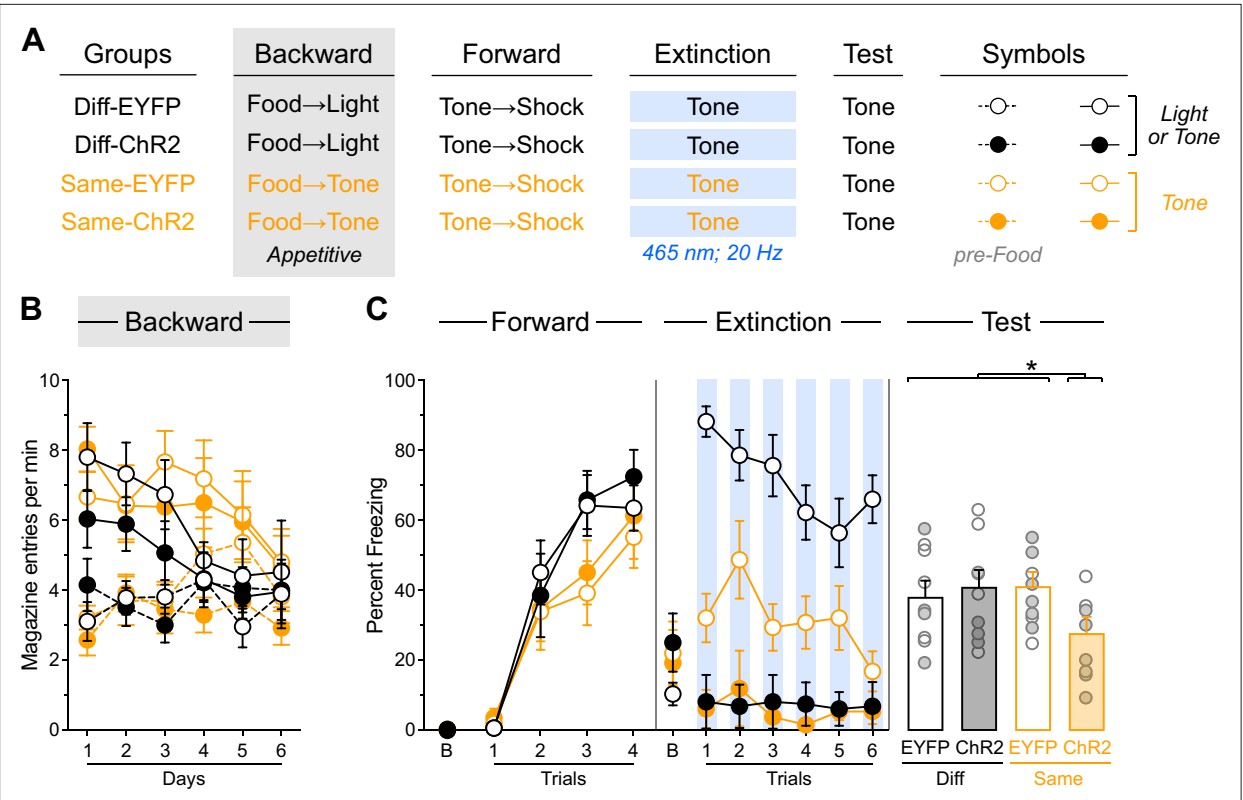

**Figure 5.** Inhibitory memories encoded within the infralimbic cortex (IL) lack motivational information. (**A**) Schematic representation of the behavioral design used in Experiment 5. The LEDs were activated in all groups during the fear extinction session. Diff-EYFP: n=4 females and 5 males; Diff-ChR2: n=4 females and 5 males; Same-EYFP: n=6 females and 3 males; Same-ChR2: n=6 females and 2 males. (**B**) Number of magazine entries during backward appetitive conditioning. (**C**) Mean percent freezing during remaining experimental stages. Data are shown as mean ± SEM. Test data include individual data points for female (filled circle) and male (open circle) rats. Asterisks denote significant effect (*$p<0.05$).

The online version of this article includes the following figure supplement(s) for figure 5:

**Figure supplement 1.** Histological and behavioral data related to *Figure 5*.

The test results revealed that backward fear conditioning produced subtractive summation. Although overall freezing was greater during light-alone presentations than during combined light and tone presentations (Period: $F_{(1,30)}$ = 11.32; $p<0.001$; $\eta^2$=0.27), this difference depended on protocol assignment (Protocol×Period: $F_{(1,30)}$ = 9.14; $p<0.01$; $\eta^2$=0.23). Simple effects analyses showed that control rats displayed equivalent freezing during both test periods ($p=0.81$). In contrast, rats with prior backward fear conditioning experience showed significantly more freezing during light-alone presentations than during combined light and tone presentations ($F_{(1,30)}$ = 20.41; $p<0.001$; $\eta^2$=0.40). No other significant differences were observed (lowest $p=0.93$), including during the baseline period (lowest $p=0.66$).

This experiment demonstrates that backward fear conditioning endowed the tone stimulus with inhibitory properties capable of reducing freezing to the excitatory light stimulus. This result is consistent with the hypothesis that backward fear conditioning protocol establishes inhibitory memories within the IL that can be retrieved and strengthened during subsequent fear extinction sessions by optogenetic stimulation of the IL during subsequent fear extinction.

## Inhibitory memories encoded within the IL lack motivational information

Experiment 5 tested whether IL memories incorporate detailed information about the biologically significant event used in backward conditioning. Rats received the same surgical procedures (*Figure 5—figure supplement 1A*) as those from Experiment 3, and identical behavioral procedures with one critical modification (*Figure 5A*): during the initial experimental stage, rats received backward

appetitive conditioning (the food outcome was delivered and 5 s after entries in the magazine where it was delivered, the stimulus was turned on). Hence, rats in each viral group were randomly assigned to one of two backward appetitive conditioning procedures: in one procedure, rats received backward conditioning with a light (Diff-EYFP and Diff-ChR2 groups), while those in the other procedure received identical conditioning using a tone (Same-EYFP and Same-ChR2 groups).

Appetitive performance is displayed as the rate of entries in the magazine where food was delivered (*Figure 5B*). Overall entries increased across training days (Day: $F_{(1,31)}$ = 5.91; $p<0.05$; $\eta^2$=0.16). Animals showed greater entries during stimulus presentations compared to pre-food periods (Period: $F_{(1,31)}$ = 79.64; $p<0.001$; $\eta^2$=0.72), with this discrimination disappearing as training progressed (Period×Day: $F_{(1,31)}$ = 20.31; $p<0.001$; $\eta^2$=0.40). An unexpected Protocol×Period interaction emerged ($F_{(1,31)}$ = 4.37; $p<0.05$; $\eta^2$=0.12). In the rats trained with the light (Diff-EYFP and Diff-ChR2 groups), overall entries were similar across virus condition (Virus: $p=0.41$). They increased across days (Day: $F_{(1,16)}$ = 5.97; $p<0.05$; $\eta^2$=0.27), were greater during stimulus presentations compared to pre-food periods (Period: $F_{(1,16)}$ = 24.02; $p<0.001$; $\eta^2$=0.76), with this discrimination disappearing as training progressed (Period x Day: $F_{(1,16)}$ = 10.46; $p<0.01$; $\eta^2$=0.40). None of these factors were influenced by viral assignment (Virus×Day, Virus×Period, Virus×Day×Period: lowest $p=0.08$). In the rats trained with the tone (Same-EYFP and Same-ChR2 groups), overall entries were similar across virus condition (Virus: $p=0.06$). They remained stable across days (Day: $p=0.25$), were greater during stimulus presentations compared to pre-food periods (Period: $F_{(1,15)}$ = 6.28; $p<0.05$; $\eta^2$=0.30), with this discrimination disappearing as training progressed (Period×Day: $F_{(1,15)}$ = 152.18; $p<0.001$; $\eta^2$=0.91). None of these factors were influenced by viral assignment (Virus×Day, Virus×Period, Virus×Day×Period: lowest $p=0.13$). No other significant differences were observed during this stage (lowest $p=0.06$). It is noteworthy that the pattern of stimulus-elicited response during backward appetitive responding differs substantially from that observed during backward fear conditioning. However, the appetitive performance observed is similar to that obtained in previous studies (*Laurent et al., 2017*; *Seitz et al., 2022*; *Laurent et al., 2018*; *Laurent and Balleine, 2015*) and should not be taken as a failure of the protocol to generate inhibitory learning. It also suggests that the unexpected interactions in this and previous stages are unlikely to have uncovered any influence on that learning.

Freezing data are presented in *Figure 5C*. Forward fear conditioning proceeded successfully, with tone-elicited freezing increasing gradually across trials (Trial: $F_{(1,31)}$ = 240.95; $p<0.001$; $\eta^2$=0.89). No other significant differences were observed (lowest $p=0.06$), including during the baseline period (lowest $p=0.26$).

The brief fear extinction session produced a gradual decline in tone-elicited freezing across trials (Trial: $F_{(1,31)}$ = 10.43; $p<0.01$; $\eta^2$=0.25), regardless of protocol assignment (Protocol: $p=0.49$). Consistent with previous experiments, IL stimulation significantly reduced tone freezing (Virus: $F_{(1,31)}$ = 178.70; $p<0.001$; $\eta^2$=0.85), and consequently, the gradual decline in freezing across trials was more pronounced in non-stimulated rats (Virus x Trial: $F_{(1,31)}$ = 12.52; $p<0.001$; $\eta^2$=0.29). Furthermore, the freezing reduction produced by IL stimulation was greater in rats that had received backward appetitive conditioning with the light (Protocol×Virus: $F_{(1,31)}$ = 23.29; $p<0.001$; $\eta^2$=0.43), as these rats displayed overall higher freezing levels than those initially trained with the tone (Protocol: $F_{(1,31)}$ = 20.35; $p<0.001$; $\eta^2$=0.40). The source of this difference remains unclear and is somewhat inconsistent with the notion that stimuli predicting the omission of an appetitive outcome (the tone in this experiment) are functionally equivalent to stimuli predicting fearful events (*Laurent et al., 2018*; *Dickinson and Dearing, 1979*). No significant differences were detected during the baseline period (lowest $p=0.23$).

The test data revealed that backward *appetitive* conditioning enables IL stimulation to facilitate subsequent *fear* extinction in a stimulus-specific manner (*Figure 5C*, *Figure 5—figure supplement 1B*). Rats with prior backward appetitive experience with the light displayed substantial and equivalent freezing levels to the tone, regardless of IL stimulation status (Diff-EYFP vs. Diff-ChR2: $p=0.28$). These freezing levels were comparable to those observed in rats that had received backward appetitive conditioning to the tone but no IL stimulation during the brief fear extinction session (Diff-EYFP +Diff-ChR2 vs. Same-EYFP: $p=0.98$). Critically, all three control groups exhibited significantly greater freezing than rats that had received both backward appetitive conditioning to the tone and IL stimulation during the brief fear extinction session (Diff-EYFP +Diff-ChR2+Same EYFP vs. Same-ChR2:

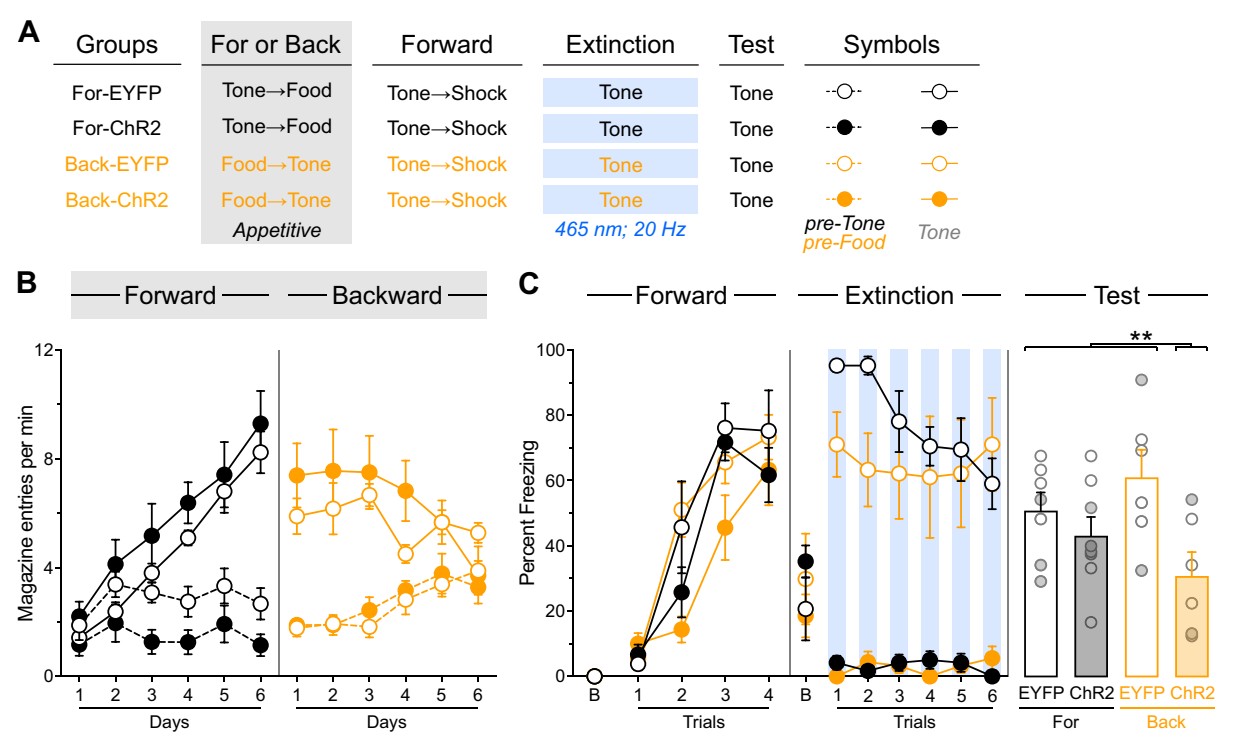

**Figure 6.** Stimulus familiarity does not enable infralimbic cortex (IL) stimulation to facilitate fear extinction. (**A**) Schematic representation of the behavioral design used in Experiment 6. The LEDs were activated in all groups during the fear extinction session. For-EYFP: n=3 females and 4 males; For-ChR2: n=4 females and 4 males; Diff-EYFP: n=2 females and 4 males; Diff-ChR2: n=2 females and 4 males. (**B**) Number of magazine entries during forward and backward appetitive conditioning. (**C**) Mean percent freezing during remaining experimental stages. Data are shown as mean ± SEM. Test data include individual data points for female (filled circle) and male (open circle) rats. Asterisks denote significant effect (**$p<0.01$).

The online version of this article includes the following figure supplement(s) for figure 6:

**Figure supplement 1.** Histological and behavioral data related to *Figure 6*.

$F_{(1,31)}$ = 6.59; $p<0.05$; $\eta^2$=0.18). No significant differences were observed during the baseline period (lowest $p$=0.26).

These findings demonstrate that backward *appetitive* conditioning enables IL stimulation to facilitate subsequent *fear* extinction to that stimulus. This pattern suggests that inhibitory memories encoded within the IL do not contain information about the motivational properties of the omitted outcome, indicating that these memory traces are motivationally neutral.

## Stimulus familiarity does not enable IL stimulation to facilitate fear extinction

The results of the previous experiments suggested that initial extinction or backward conditioning generates an inhibitory memory that can be retrieved and strengthened through IL stimulation during subsequent fear extinction. However, an alternative explanation is that the initial training session simply increased familiarity with the stimulus, and this enhanced familiarity—rather than inhibitory memory formation—enabled the facilitative effects of IL stimulation. Experiment 6 tested this alternative. Rats received the same surgical procedures (*Figure 6—figure supplement 1A*) and behavioral protocols as the previous experiment with one critical modification (*Figure 6A*). Instead of using backward appetitive conditioning to the light as the control condition, we replaced these control rats with animals that underwent forward appetitive conditioning to the tone (For-EYFP and For-ChR2). We contrasted the performance of these rats with that of rats that received backward appetitive conditioning to the tone (Back-EYFP and Back-ChR2).

Appetitive data are shown in *Figure 6B*. Forward appetitive conditioning was successful; both For-EYFP and For-ChR2 groups showed a gradual increase in magazine entries across training days (Day: $F_{(1,13)}$ = 61.28; $p<0.001$; $\eta^2$=0.83). Animals made significantly more entries during tone presentations

compared to pre-tone periods (Period: $F_{(1,13)}$ = 65.68; $p<0.001$; $\eta^2$=0.84), and this discrimination became increasingly pronounced as training progressed (Period×Day: $F_{(1,13)}$ = 187.24; $p<0.001$; $\eta^2$=0.94). No other significant effects were observed during this phase (lowest $p$=0.15). Backward appetitive conditioning proceeded as expected (Back-EYFP and Back-ChR2). While overall entries did not increase across days (Day: $p$=0.90), animals showed greater entries during tone presentations than during pre-food periods (Period: $F_{(1,10)}$ = 97.53; $p<0.001$; $\eta^2$=0.92). This temporal discrimination pattern weakened over training sessions (Period×Day: $F_{(1,10)}$ = 22.86; $p<0.001$; $\eta^2$=0.70), with no other significant effects observed (lowest $p$=0.25).

Freezing data are presented in *Figure 6C*. Forward fear conditioning proceeded successfully, with tone-elicited freezing increasing gradually across trials (Trial: $F_{(1,23)}$ = 177.02; $p<0.001$; $\eta^2$=0.44). An unexpected but significant effect of viral assignment emerged ($F_{(1,23)}$ = 5.99; $p<0.05$; $\eta^2$=0.21), though no other significant differences were found during this stage, including during the baseline period (lowest $p$=0.07).

The brief fear extinction session produced a gradual decline in tone-elicited freezing across trials (Trial: $F_{(1,23)}$ = 7.10; $p<0.05$; $\eta^2$=0.25). As expected, IL stimulation significantly reduced freezing responses (Virus: $F_{(1,23)}$ = 248.83; $p<0.001$; $\eta^2$=0.92). Consequently, the gradual decline in freezing was more pronounced in non-stimulated animals (Virus×Trial: $F_{(1,23)}$ = 8.06; $p<0.01$; $\eta^2$=0.26). Interestingly, rats that had received forward appetitive conditioning showed a more pronounced decline in freezing (Protocol×Trial: $F_{(1,23)}$ = 9.02; $p<0.01$; $\eta^2$=0.28), resulting in a significant three-way interaction (Protocol×Virus×Trial: $F_{(1,23)}$ = 5.37; $p<0.05$; $\eta^2$=0.19). While the source of these differences remains unclear, they may reflect delayed fear conditioning to the appetitive inhibitor (Back-EYFP group) combined with enhanced fear conditioning to the appetitive excitor (For-EYFP group), as suggested by previous research (*Laurent et al., 2018*; *Dickinson and Dearing, 1979*; *Dickinson, 1976*). No significant differences were observed during the baseline period (lowest $p$=0.26).

The test data provided clear evidence that backward appetitive conditioning, not stimulus familiarity, enables IL stimulation to facilitate fear extinction (*Figure 6C*, *Figure 6—figure supplement 1B*). Rats with prior forward appetitive experience to the tone displayed substantial and equivalent freezing levels to the tone, regardless of IL stimulation status (For-EYFP vs. For-ChR2: $p$=0.40). These freezing levels were comparable to those observed in rats that had received backward appetitive conditioning to the tone but no IL stimulation during the brief fear extinction session (For-EYFP+For-ChR2 vs. Back-EYFP: $p$=0.11). Critically, all three control groups exhibited significantly greater freezing than rats that had received both backward appetitive conditioning to the tone and IL stimulation during the brief fear extinction session (For-EYFP+For-ChR2+Back EYFP vs. Back-ChR2: $F_{(1,31)}$ = 6.81; $p<0.05$; $\eta^2$=0.23). No significant differences were observed during the baseline period (lowest $p$=0.41).

These findings demonstrate that stimulus familiarity per se does not enable IL stimulation to facilitate fear extinction. Instead, the results support our original hypothesis: initial extinction or backward conditioning generates a specific inhibitory memory that can be retrieved and strengthened by IL stimulation during subsequent fear extinction to the same stimulus.

## Prior experience with appetitive extinction enables IL stimulation to facilitate subsequent backward fear conditioning

Experiment 7 investigated whether the facilitative effects of IL stimulation are limited to fear extinction sessions or can be observed when stimulation occurs during other types of learning that involve inhibition. Specifically, we examined whether these facilitative effects could also be uncovered during backward fear conditioning. Rats received bilateral IL infusions of either EYFP or ChR2 virus and were implanted with fiber-optic cannulae targeting the IL (*Figure 7—figure supplement 1A*). All animals underwent the same behavioral protocol (*Figure 7A*) using the tone as the only stimulus. The sequence of behavioral stages was: forward appetitive conditioning, appetitive extinction, forward fear conditioning, backward fear conditioning, and test. LED activation occurred during 6 of the 8 tone presentations during the backward fear conditioning phase.

Appetitive data are shown in *Figure 7B*. Forward appetitive conditioning was successful, with overall entries increasing gradually across days (Day: $F_{(1,17)}$ = 106.67; $p<0.001$; $\eta^2$=0.86). Animals made significantly more entries during tone presentations compared to pre-tone periods (Period: $F_{(1,17)}$ = 19.50; $p<0.001$; $\eta^2$=0.53), and this discrimination became increasingly pronounced as training progressed (Period×Day: $F_{(1,13)}$ = 144.51; $p<0.001$; $\eta^2$=0.89). No other significant effects were

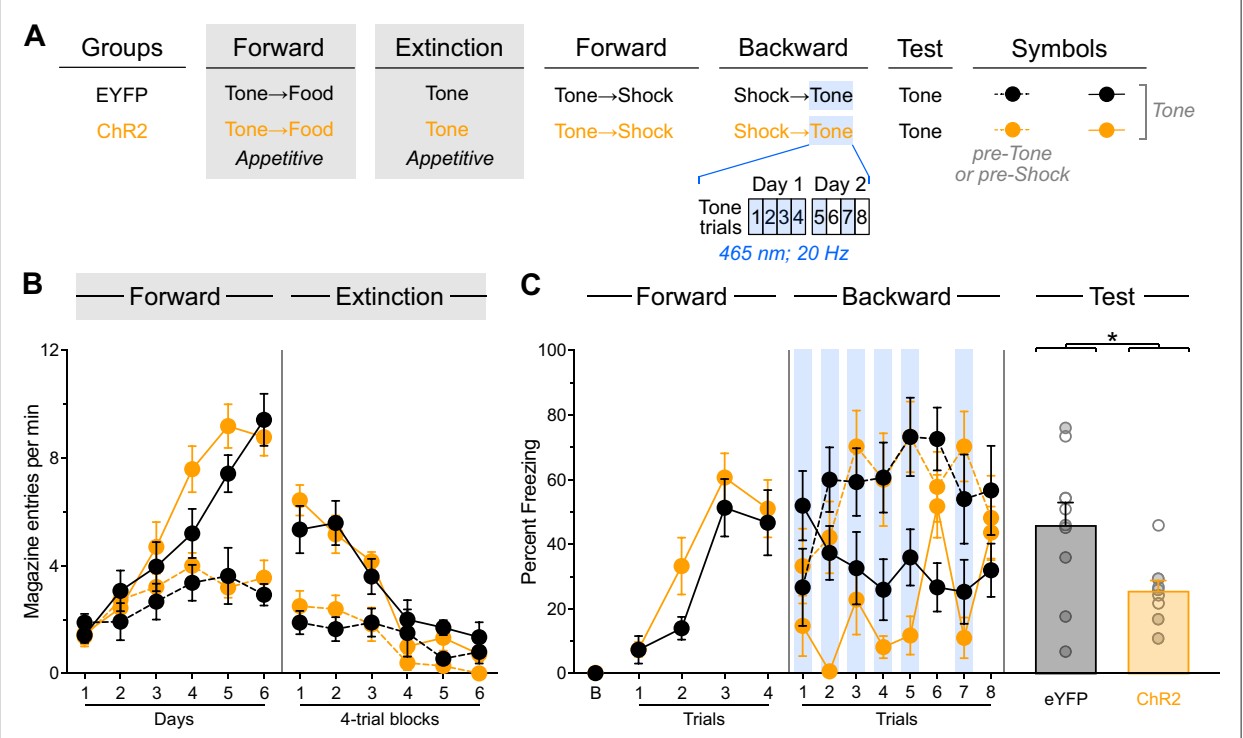

**Figure 7.** Prior experience with appetitive extinction enables infralimbic cortex (IL) stimulation to facilitate subsequent backward fear conditioning. (**A**) Schematic representation of the behavioral design used in Experiment 7. The LEDs were activated in all groups during backward fear conditioning. EYFP: n=5 females and 5 males; ChR2: n=5 females and 4 males. (**B**) Number of magazine entries during forward appetitive conditioning and appetitive extinction. (**C**) Mean percent freezing during remaining experimental stages. Data are shown as mean ± SEM. Test data include individual data points for female (filled circle) and male (open circle) rats. Asterisks denote significant effect (*p<0.05).

The online version of this article includes the following figure supplement(s) for figure 7:

**Figure supplement 1.** Histological and behavioral data related to *Figure 7*.

observed during this phase (lowest $p$=0.15). Appetitive extinction produced a gradual decrease in overall entries (Trial: $F_{(1,17)}$ = 105.63; $p$<0.001; $\eta^2$=0.92). Although entries were greater during tone presentations compared to pre-tone periods (Period: $F_{(1,17)}$ = 55.67; $p$<0.001; $\eta^2$=0.77), this discrimination became progressively smaller (Period×Trial: $F_{(1,17)}$ = 35.53; $p$<0.001; $\eta^2$=0.68).

Freezing data are presented in *Figure 7C*. Forward fear conditioning proceeded successfully, with tone-elicited freezing increasing across trials (Trial: $F_{(1,17)}$ = 52.83; $p$<0.001; $\eta^2$=0.76), regardless of viral assignment ($p$=0.95). No other significant differences were found during this stage, including during the baseline period (lowest $p$=0.026). Backward fear conditioning proceeded as expected, with overall freezing greater during the pre-shock periods compared to stimulus presentations (Period: $F_{(1,17)}$ = 43.54; $p$<0.001; $\eta^2$=0.72). Critically, tone-elicited freezing was reduced by LED activation (LED: $F_{(1,17)}$ = 11.98; $p$<0.01; $\eta^2$=0.41), but this reduction was only observed in ChR2 rats (LED×Virus: $F_{(1,17)}$ = 22.25; $p$<0.001; $\eta^2$=0.57). No other significant differences were observed during this stage (lowest $p$=0.06).

The test data revealed that IL stimulation during backward fear conditioning successfully reduced subsequent freezing (*Figure 7C*, *Figure 7—figure supplement 1B*): ChR2 rats froze less than EYFP rats (Virus: $F_{(1,17)}$ = 6.67; $p$<0.05; $\eta^2$=0.28). This finding demonstrates that appetitive extinction enables IL stimulation to facilitate subsequent backward fear conditioning. This indicates that the facilitative effects of IL stimulation are not limited to fear extinction sessions but can be observed when stimulation occurs during other procedures that involve inhibitory learning. This pattern suggests that procedures generating inhibition create a memory within the IL that can be accessed during other inhibitory learning situations.

## Discussion

The present experiments explored how the IL encodes inhibitory memories by investigating the conditions that promote their retrieval. Initial ex-vivo electrophysiological recordings confirmed that optogenetic stimulation successfully increased activity in IL neurons. We then found that prior experience with fear extinction enables IL stimulation to facilitate subsequent fear extinction. The facilitative effects of IL stimulation were not limited to fear extinction experience, as prior exposure to backward fear conditioning also enabled IL stimulation to enhance later fear extinction. These effects were stimulus-specific, occurring only for the stimulus that had undergone backward fear conditioning, a procedure that was found to generate the characteristics of inhibitory learning. Remarkably, backward appetitive conditioning also allowed IL stimulation to facilitate later fear extinction, suggesting that inhibitory memories in the IL lack information about the motivational circumstances present during their encoding. Additional experiments ruled out stimulus familiarity as an explanation for the facilitative effects of IL stimulation and demonstrated that IL-mediated facilitation can be observed in procedures other than fear extinction. Together, these findings provide novel insights into how the brain processes and stores inhibitory information and broaden our understanding of IL function in adaptive behavior.

Throughout all experiments, IL stimulation produced an immediate suppression of the freezing responses elicited by the conditioned stimulus. This effect aligns with the existing literature (*Do-Monte et al., 2015*; *Kim et al., 2016*; *Milad et al., 2004*; *Vidal-Gonzalez et al., 2006*; *Lingawi et al., 2018*; *Lingawi et al., 2017*; *Thompson et al., 2010*; *Chang and Maren, 2011*), but it did not predict the long-term consequences of the stimulation. This dissociation between immediate and long-term effects was evident in Experiment 1. While IL stimulation suppressed fear responses during both the first and second extinction sessions, it only facilitated the fear inhibition produced by the second session, not the first. This result confirmed that prior experience with fear extinction creates the conditions necessary for IL stimulation to enhance subsequent extinction learning (*Chang and Maren, 2011*; *Lingawi et al., 2017*). Critically, the finding in Experiment 2 that backward fear conditioning experience also enabled IL stimulation to facilitate subsequent fear extinction demonstrated that IL engagement extends beyond situations where stimuli are encountered in isolation (e.g. extinction) to encompass protocols believed to generate inhibitory learning through different mechanisms (*Sosa, 2024*). This finding was replicated in Experiment 3, which also showed that the facilitative effects of IL stimulation were not due to background context fear and only emerged when the same stimulus undergoes both backward fear conditioning and subsequent fear extinction, indicating that backward fear conditioning generates stimulus-specific memories within the IL. These memories were found to be inhibitory, as Experiments 2 and 3 revealed retarded fear conditioning to the backward-trained stimulus, while Experiment 4 found that such a stimulus reduced the fear-eliciting capacity of a separately fear-conditioned stimulus, both of which are hallmark properties of inhibitory stimuli (*Rescorla, 1969*). Collectively, these findings support the hypothesis that fear extinction and backward fear conditioning generate stimulus-specific inhibitory memories that can be subsequently reactivated during a fear extinction session to facilitate the inhibition produced by that session.

Experiment 5 asked about the motivational characteristics of inhibitory memories encoded within the IL. It found that prior experience with *appetitive* backward conditioning enabled IL stimulation to facilitate *fear* extinction in a stimulus-specific manner. This suggests that inhibitory memories stored in the IL are stripped of the motivational information that is present during their initial encoding, a suggestion that is consistent with the earlier observation that *appetitive* extinction enables IL stimulation to facilitate subsequent *fear* extinction (*Lingawi et al., 2018*). Experiment 6 addressed the alternative hypothesis that IL stimulation became effective in all experiments simply because prior experiences enhanced stimulus familiarity. The results demonstrated that prior experience with backward appetitive conditioning, but not forward appetitive conditioning, enabled IL stimulation to facilitate subsequent fear extinction. Since the backward and forward protocols provided similar exposure to the target stimulus, this finding rules out familiarity as the determining factor in IL engagement and supports the view that the critical determinant is the capacity of prior experiences to produce inhibitory learning. Finally, Experiment 7 sought to demonstrate that IL-encoded inhibitory memories can be retrieved during inhibitory procedures other than fear extinction, thereby expanding the scope of IL function beyond that of fear extinction. It was found that prior experience with appetitive extinction enabled IL stimulation to facilitate the development of fear inhibition during backward

fear conditioning. This finding generalizes the effects of IL stimulation to another inhibitory protocol, providing further evidence that the IL generally contributes to inhibitory learning even in different motivational contexts.

The present experiments aimed to uncover the general role of the IL in encoding inhibitory memories. Accordingly, they employed a non-selective optogenetic approach that stimulated all IL neurons. The results reveal an important difference with studies that employed cell-specific stimulation. Although selective stimulation of IL pyramidal neurons has been found to facilitate initial extinction learning (*Do-Monte et al., 2015*; *Chen et al., 2021*), this facilitation was absent in the present experiments, which instead replicated previous work that used a pharmacological IL stimulation approach (*Lingawi et al., 2018*; *Lingawi et al., 2017*). These contrasting effects extend previous findings showing that extinction retrieval is disrupted by broad optogenetic silencing of the IL (*Kim et al., 2016*) but is left unaffected by selective silencing of IL pyramidal neurons (*Do-Monte et al., 2015*). Together, these discrepancies highlight the critical role that IL interneurons may play in shaping both the encoding and expression of inhibitory memories, which is consistent with recent research uncovering an important function for these interneurons in extinction learning (*Ho et al., 2025*; *Marek et al., 2018*; *Binette et al., 2023*; *Koppensteiner et al., 2019*). Future studies should, therefore, identify the specific neuronal populations that are responsible for the effects observed in the present experiments. Additionally, it will be necessary to examine how these effects are modulated by other brain regions that have been identified as important regulators of inhibitory learning. These include the amygdala (*Do-Monte et al., 2015*; *Laurent et al., 2008*; *Laurent et al., 2008*), the hippocampus (*Hassell et al., 2025*; *Marek et al., 2018*), the thalamus, locus coeruleus (*Bayer et al., 2025*; *Do-Monte et al., 2010*; *Mueller et al., 2008*; *Uematsu et al., 2017*), the orbitofrontal cortex (*Lay et al., 2020*; *Rodriguez-Romaguera et al., 2015*), and the lateral habenula (*Matsumoto and Hikosaka, 2007*; *Laurent et al., 2017*; *Sosa et al., 2021*).

Other aspects of the present data merit consideration. First, despite our efforts to match group performance during training, post-mortem exclusions resulted in unexpected group differences in some experiments. However, these differences were typically transient or unable to account for the final experimental outcomes. Moreover, the key findings presented here are replicated across several experiments within the current study (e.g. prior experience with backward fear or appetitive conditioning) or reproduced previous work (prior experience with fear or appetitive extinction; *Lingawi et al., 2018*; *Lingawi et al., 2017*). Therefore, considerable evidence attests to the reliability and replicability of the findings reported. A second consideration relates to the mechanisms underlying the IL facilitatory effects observed here. Future research will be needed to determine whether IL stimulation enhances the subsequent retrieval of inhibitory memories or accelerates the relearning of these memories, as either process could account for the improved performance observed in our experiments. Finally, additional studies will be needed to establish whether the characteristics of IL function described here apply equally to both sexes. Although our experiments included both female and male rats, they lacked sufficient statistical power to detect sex differences. Given the substantial evidence for sex differences in extinction learning and related behaviors (*Tang and Graham, 2020*), this remains an important question for future investigation.

Collectively, the present findings reveal fundamental principles governing how the IL encodes and retrieves inhibitory memories. Our experiments demonstrate that the IL stores inhibitory memories that are extremely flexible since they can be retrieved and used across many inhibitory procedures and distinct motivational contexts. This motivational independence offers a sophisticated neural mechanism that allows the application of previously learned inhibitory relationships to novel circumstances. Whether inhibitory memories operate entirely outside motivational control or whether this autonomy specifically characterizes the components that are encoded in the IL remains unknown, though previous work showing that inhibitory stimuli can modulate learning and behavior, regardless of primary motivational states suggests that inhibitory memories may broadly escape motivational regulation (*Laurent et al., 2018*). Regardless, the stimulus specificity of the reported effects reveals precision in IL memory encoding, restricting retrieval to appropriate situations and balancing for the generalizability of retrieval provided its motivational independence. By ruling out alternative explanations, such as context fear or stimulus familiarity, our experiments also provide evidence that the IL is primarily concerned with inhibitory learning rather than with more basic associative processes. Thus, the IL emerges as a critical hub for the flexible application of

inhibitory knowledge that enables adaptive responses in the face of ever-changing environmental circumstances.

## Materials and methods

### Subjects

Subjects were 248 experimentally naïve female and male Long-Evans rats obtained from the breeding facility at the University of New South Wales (Sydney, Australia). The rats were at least 8 weeks old and weighed 300–500 grams at the beginning of each experiment. Efforts were made to match group performance across the training stages and to allocate an equal number of female and male rats in each group. However, these efforts were hampered by animal availability and the necessary exclusions following post-mortem assessments of viral spread and fiber-optic cannula placement. The final number of female and male rats did not provide sufficient power to analyze an effect of sex. Therefore, all test data present individual data for female and male rats. Rats were housed in transparent intraventilated cages (Tecniplast Australia) with their littermates (up to four rats per cage) in a climate-controlled colony room maintained on a 12 hr light-dark cycle (lights on at 7:00 am). Experimental procedures were conducted during the light phase (8:00 am to 6:00 pm). Water and standard lab chow were available ad libitum except in experiments involving the delivery of food outcomes. In these experiments, rats were food restricted to maintain them at ~90% of their ad libitum body weight. Food restriction started 3–4 days before the start of the experiments and was maintained throughout the behavioral procedures. All rats were handled daily 3–4 days before the start of the experiments. All the animals were handled according to approved Animal Care and Ethics Committee (ACEC) protocols of the University of New South Wales. The experimental protocols were approved by the UNSW ACEC (Permit Number: 21/28B). All surgery was performed under isoflurane anesthesia, and every effort was made to minimize suffering.

### Viruses

The following viruses were used: AAV5-hSyn-eYFP (**EYFP**; University of North Carolina Vector Core; Chapel Hill, NC, USA) and AAV5-hSyn-hChR2(H134R)-EYFP (**ChR2**; Addgene; Watertown, MA, USA; #26973).

### Surgery

Stereotaxic surgery was conducted under isoflurane gas anesthesia (0.8 L/min; induction at 4–5%, maintenance 2–2.5%). Animals were placed in a stereotaxic frame (Kopf Instruments, Tujunga, CA, USA), the surgical area was shaved, and betadine iodine antiseptic solution was applied. At the incision site, bupivacaine hydrochloride (0.5%; 0.1–0.2 mL) was injected subcutaneously. In the lower flank, Metacam (1 mg/kg) was injected subcutaneously. An incision was made along the midline to expose the skull, the membrane on top of the skull was cleared, and the skull was adjusted to align bregma and lambda on the same horizontal plane. Small holes were drilled into the skull above the infralimbic cortex (IL) using the following coordinates A/P: +2.45, M/L: ±2.0, D/V: –5.1 at 15° angle. An Infuse/Withdraw Pump (Standard Infuse/Withdraw Pump 11 Elite; Harvard Apparatus, Holliston, MA, USA) in combination with glass 1 µL syringes (86200; Hamilton Company, Reno, NV, USA) were used to perform viral infusions. 0.5 µL of the virus (EYFP or ChR2) was infused in each hemisphere at a rate of 0.1 µL/min. Following each infusion, the needle was left in position for 5–10 min to allow for diffusion. Fiber-optic cannulae were then bilaterally implanted 0.5 mm above and fixed with jeweler's screws and dental cement. Animals then received intraperitoneal injections of sodium chloride (0.9%, 2 mL) and procaine penicillin (Ilium Benicillin; 300 mg/kg) were placed on a heat mat for recovery. Once alert and responsive, animals were returned to their home cages and monitored daily as they recovered for 3 weeks to allow for viral expression before behavioral procedures commenced.

### Behavioral apparatus

Training and testing were conducted in 8 identical operant chambers (ENV-007-VP; L 29.53 x W 23.5 x H 27.31 cm; MED Associates, Fairfax, VT, USA) enclosed within light- and sound-attenuating cabinets (ENV-018MD; L 59.69 x W 40.64 x H 55.88 cm; MED Associates, Fairfax, VT, USA). The chambers consisted of two stainless-steel side walls, three clear polycarbonate walls and ceiling, and a

stainless-steel grid floor. Each chamber was equipped with a speaker that delivered a sound card-generated (ANL-926; MED Associates, Fairfax, VT, USA) 90 dB, 1000 Hz tone. LED lights mounted on the back wall of each sound-resistant cabinet were used to deliver a flashing light stimulus presented at 2 Hz (~8 lx). The duration of the tone and flashing stimuli was set at 30 s in all behavioral procedures. The grid floor of each chamber was connected to a shock generator (ENV-410C; MED Associates, Fairfax, VT, USA) that delivered a 0.5 mA, 0.5 s shock. Grain pellets (F0165; Bio-Serv Biotechnologies, Flemington, NJ, USA) could be delivered through a food magazine in one wall of the chamber. Head entries into the magazine were detected via an infrared beam that crossed the magazine opening. The ceiling of the chambers contained a hole that enabled the connection of the patch cable between the LED and each freely moving animal. Training and testing sessions were programmed and controlled by computers external to the testing rooms using MED-PC V software (MED Associates, Fairfax, VT, USA) which also recorded experimental data from each session. All cabinets were fitted with cameras (IPC-K35AP; Dahua, Artarmon, NSW, Australia) to record and view real-time chamber activity.

## Optogenetic equipment

Fiber-optic cannulas (10 mm; RWD Life Science Co., Guangdong, China) consisted of a ceramic ferrule from which a silica/polymer flat-tip 400 nM core fibre extended. Fiber-optic patch cords (custom; Doric Lenses, Québec, QC, Canada) were enclosed in a flexible cladding that consisted of one transparent fiber that split into two smaller fibres for attachment to the implanted fiber-optic cannulae. Delivery of the LED light was controlled through a TTL adapter which converted 28 V DC output to a TTL transition (MED Associates, Fairfax, VT, USA). The adaptor was connected to an LED driver (Doric Lenses, Québec, QC, Canada) and Connectorized LED light source (Doric Lenses, Québec, QC, Canada) with a fiber-optic rotary joint for attachment to the fiber-optic patch cords. LED light was delivered as blue light (465 nm) and measured at least 10 mW at the cannula tip using a photometer (PM200; ThorLabs, Newton, NJ, USA) when the LED was switched on.

## Behavioral procedures

All experiments employed parameters similar to those from our previous studies (*Laurent et al., 2022*; *Lingawi et al., 2018*; *Laurent et al., 2018*; *Lingawi et al., 2017*).

### Experiment 1
#### Day 1: Pre-exposure
All rats received pre-exposure to the experimental chambers and tone stimulus during a 30 min session. The tone was presented twice: first after 11 min, then again 11 min later, with rats removed 7 min after the second presentation. Patch cords were connected to fiber-optic cannulae and LEDs throughout this session, though LEDs remained inactive.

#### Day 2: Forward fear conditioning
All rats underwent forward fear conditioning with four tone-shock pairings. The tone stimulus co-terminated with a single shock, with pairings separated by pseudorandom inter-trial intervals (ITI) of 3–5 min (4 min average). Total session duration was 20 min.

#### Day 3: Standard fear extinction
Rats in the ReExt-OFF and ReExt-ON groups received their first extinction session, consisting of 16 tone-alone presentations with variable ITI of 3–5 min over 58 min. Rats in the Ext-OFF and Ext-ON groups were handled only.

#### Day 4: Second fear conditioning
All rats received an additional forward fear conditioning session.

#### Day 5: Brief fear extinction
All rats underwent a shortened extinction session with eight tone-alone presentations and variable ITI of 3–5 min, lasting 38 min total. This represented the first extinction session for Ext-OFF and Ext-ON groups, but the second for ReExt-OFF and ReExt-ON groups. Critically, patch cords were attached to

fiber-optic cannulae and LEDs during this session, with LEDs activated during tone presentations for the Ext-ON and ReExt-ON groups only.

### Day 6: Test
All rats received eight tone-alone presentations with variable ITIs of 3–5 min over a 38 min session.

## Experiment 2
### Day 1: Pre-exposure
All rats underwent pre-exposure sessions using previously described methods.

### Days 2-6: Backward fear conditioning
Rats in the Back-EYFP and Back-ChR2 groups received five daily backward fear conditioning sessions with the tone stimulus. Each session consisted of four shock-tone trials, where shocks were delivered 5 s before tone presentation. A variable ITI of 3–5 minwere used, with each session lasting 20.5 min. Rats in the No-EYFP and No-ChR2 groups were handled only during this period.

### Days 7-8: Context extinction
All rats underwent two daily 30 min exposure sessions in the conditioning chambers without stimulus presentation. During these sessions, rats were habituated to patch cord attachment using the pre-exposure protocol.

### Day 9: Forward fear conditioning
All rats received forward fear conditioning to the tone using previously described methods.

### Day 10: Brief fear extinction
All rats underwent a shortened extinction session to the tone following previously described protocols. Patch cords were connected to fiber-optic cannulae and LEDs throughout this session, with LEDs activated during tone presentations for all experimental groups.

### Day 11: Test
All rats were tested for conditioned fear responses to the tone using previously described procedures.

## Experiment 3
Experiment 3 followed the same protocol as Experiment 2, with one key modification: control groups (Diff-EYFP and Diff-ChR2) received backward fear conditioning to a light stimulus, while experimental groups (Same-EYFP and Same-ChR2) received backward fear conditioning to the tone stimulus.

## Experiment 4
### Days 2-6: Backward fear conditioning
The Back group received backward fear conditioning to the tone stimulus following previously described methods. The No group underwent identical procedures with tone presentations excluded.

### Days 7-8: Context extinction
All rats received two daily 30 min sessions of chamber exposure without stimulus presentation.

### Day 9: Forward fear conditioning
All rats underwent forward fear conditioning using previously described protocols, substituting the light stimulus for the tone stimulus.

### Day 10: Test
All rats were tested for conditioned fear responses to both the light alone and the light-tone compound stimulus. Each stimulus type was presented 4 times in pseudorandom order (ABBABAAB; with A=light and B=light+tone) with variable ITI of 3–5 min over a 38 min session.

### Experiment 5

Experiment 5 followed the same protocol as Experiment 3, with backward appetitive conditioning replacing backward fear conditioning.

#### Days 2-7: Backward appetitive conditioning

Over six daily sessions, rats received conditioning with either the light stimulus (Diff-EYFP and Diff-ChR2 groups) or the tone stimulus (Same-EYFP and Same-ChR2 groups). Each session consisted of six trials with variable ITIs of 3–5 min, lasting approximately 60 min total. Each trial began with the delivery of two grain pellets. Upon detection of a food magazine entry, the light or tone stimulus was activated 5 s later and continued for 30 s before terminating.

### Experiment 6

Experiment 6 followed the same protocol as Experiment 5, with one key modification: on days 2–7, control rats (For-EYFP and For-ChR2 groups) received forward appetitive conditioning with the tone stimulus instead of backward appetitive conditioning with the light stimulus.

#### Days 2-7: Forward appetitive conditioning

The same parameters as backward appetitive conditioning were used, except that two grain pellets were delivered immediately after tone stimulus presentation.

### Experiment 7

Rats underwent the following sequence: pre-exposure, forward appetitive conditioning, appetitive extinction, forward fear conditioning, backward fear conditioning, and a final test. All stages employed the tone stimulus using previously described parameters with minor modifications: backward fear conditioning occurred over only two days and was conducted with LED activation during all tone presentations on the first day but only during the first and third tone presentation on the second day. Appetitive extinction involved 2 daily sessions with 15 tone-alone presentations and variable ITIs of 3–5 min.

## Tissue preparation, histology, and microscopy

Rats were euthanised with an intraperitoneal injection of sodium pentobarbital (1 mL) and transcardially perfused with 4% paraformaldehyde (PFA) in 0.1 M sodium phosphate buffer (PBS, pH 7.5; 400 mL for rats). Brains were immediately removed and placed in individual specimen jars containing PFA and stored at 4 °C overnight for post-fixation. Brains were sliced into 40 µm coronal or sagittal sections in 0.1 M Tris-buffered saline (TBS, 0.25 M Tris, 0.5 M NaCl, pH 7.5) using a vibratome (VT1200; Leica Microsystems, North Ryde, NSW, Australia) and stored at –20 °C in cryoprotective solution containing 30% ethylene glycol, 30% glycerol, and 0.1 M PBS. Free-floating sections were selected in TBS to contain the region of interest and washed three times for 10 min each in TBS solution. Next, sections were mounted onto glass slides using Vectashield hardset antifade mounting medium (H-1400; Vector Laboratories, Newark, CA, USA).

Imaging of the mounted brain sections was performed on a confocal microscope (BX61W1 or FV10i; Olympus, Shinjuku, Tokyo, Japan) using Fluoview software (FV1000; Olympus, Shinjuku, Tokyo, Japan). Acquired images were used to assess the spread of viral infusions and placement of fiber-optic cannulae. Based on the latter, the following numbers of rats were excluded from the statistical analyses: Experiment 1 (n=3), Experiment 2 (n=11), Experiment 3 (n=6), Experiment 5 (n=3), Experiment 6 (n=3), and Experiment 7 (n=3).

## Electrophysiological recordings

### Brain slice preparation

Rats were euthanised under isoflurane gas anaesthesia (4% in air) four weeks following bilateral IL infusion of the ChR2 virus. Brains were immediately removed and sliced using a vibratome (VT1200; Leica Microsystems, North Ryde, NSW, Australia) in ice-cold oxygenated sucrose buffer solution (consisting of [in mM]: 241 sucrose, 28 $NaHCO_3$, 11 glucose, 1.4 $NaH_2PO_4$, 3.3 KCl, 0.2 $CaCl_2$, 7 $MgCl_2$). Coronal brain slices (300 µm thickness) containing the IL were sampled and maintained at 33 °C in a submerged

chamber containing physiological saline (consisting of [in mM]: 124 NaCl, 2.5 KCl, 1.25 NaH$_2$PO$_4$, 1 MgCl$_2$, 1 CaCl$_2$, 10 glucose, and 26 NaHCO$_3$) and equilibrated with 95% O$_2$ and 5% CO$_2$.

## Recordings

After equilibration for 1 hr, slices were transferred to a recording chamber and neurons visualized under an upright microscope (BX50WI, Olympus) using differential interference contrast (DIC) Dodt tube optics and EYFP fluorescence, and superfused continuously (1.5 ml/min) with oxygenated physiological saline at 33 °C. Cell-attached and whole-cell patch-clamp recordings were made using electrodes (2–5 MW) containing internal solution (in mM): 115 K gluconate, 20 NaCl, 1 MgCl$_2$, 10 HEPES, 11 EGTA, 5 Mg-ATP, and 0.33 Na-GTP, 5 phosphocreatine di(tris) salt (#P1937, Sigma-Aldrich), pH 7.3, osmolarity 285–290 mOsm/L. Biocytin (0.1%; #B4261, Sigma) was added to the internal solution for marking the sampled neurons during recording. Data acquisition was performed with a Multiclamp 700B amplifier (Molecular Devices), connected to a PC computer and interface ITC-18 (Instrutech). Liquid junction potentials of –10 mV were not corrected. In cell-attached and then whole-cell current-clamp modes, membrane current and potentials were sampled at 5 kHz (low pass filter 2 kHz, Axograph X, Axograph).

## LED stimulation

Transfected neurons with ChR2 in the IL were illuminated by a pulse train of LED light 470 nm, 5 ms pulse, 20 Hz, 0.5 mW, for 31 s train in cell-attached mode, and then for 2.5 s train in whole-cell mode (ThorLabs, Newton, NJ, USA) onto the brain slice under the 40 X water-immersion objective.

## Post-hoc histological analysis

Immediately after physiological recording, brain slices containing biocytin-filled neurons were fixed overnight in 4% paraformaldehyde/0.16 M phosphate buffer (PB) solution, rinsed and then placed in 0.5% Triton X-100/PB for 3 days to permeabilize cells. Slices were then incubated in primary antibody for 3 days, chicken anti-GFP (GFP-1020, 1:1000, Aves Labs) to enhance EYFP signal, plus 2% horse serum and 0.2% Triton X-100/PB at 4 °C. After rinsing, this was followed by a one-step overnight incubation at 4 °C in a fluorescent avidin plus a secondary antibody, ExtrAvidin Cy3 (E4142, 1:1000, Sigma), and donkey anti-chicken Alexa 488 conjugated IgY (703-545-155, 1:1000, Jackson ImmunoResearch Inc). Stained slices were rinsed, mounted onto glass slides, dried, and coverslipped with Fluoromount-G mounting medium (00-4958-02, Invitrogen). A 2D projection was later obtained from a collated image stack using confocal laser scanning microscopy (Fluoview FV1000, BX61WI microscope, Olympus).

## Statistical analyses

Two rats from Experiment 1 (group ReExt-ON) were excluded from the final analyses because they failed to display freezing during the first forward fear conditioning and extinction stages. The data presented met the assumptions of the statistical test used. The main measures were percent freezing and number of magazine entries per minute. Freezing was rated in a time-sampling manner and judged as either freezing or not freezing every 2 s by a trained observer blind to the subjects' group assignment. Percent freezing was calculated for three periods: baseline was the period prior to the onset of the first stimulus presentation, pre-shock was the period during the 30 s prior to the onset of each shock during backward fear conditioning, and stimulus was the period during the 30 s stimuli (tone, light, or light+tone). The number of magazine entries per minute was also calculated for three periods: pre-stimulus was the period during the 30 s prior to the onset of each stimulus presentation, pre-food was the period during the 30 s prior to the onset of each 2-pellet delivery during backward appetitive conditioning, and stimulus was the period during the 30 s stimuli (tone or light).

The difference between groups were analyzed using a planned orthogonal contrast procedure controlling the per contrast error rate (*Hays, 1963*). We first consider the 4-group design experiments (Experiments 1, 2, 3, 5, and 6). All training stages included a first contrast for behavioral procedure (e.g. Ext vs. ReExt or No vs. Back), a second contrast for either LED (ON vs. OFF in Experiment 1) or Virus (EYFP vs. ChR2) and a third contrast testing for the interaction between these two contrasts. For the test stage, the contrasts were informed by our a priori hypotheses. The first contrast tested for differences between the two behavioral control groups (e.g. Ext-EYFP vs. Ext-ChR2, or No-EYFP

vs. No-ChR2). The second contrast tested for differences between the two previous control groups and the behavioral experimental group in the control LED or Virus condition (e.g. ReExt-OFF, or Back-EYFP). The final contrast tested for differences between the previous three groups and the critical behavioral group in the LED ON or Virus ChR2 condition. A single between-subject contrast was used in the 2-group design experiments (Experiments 4 and 7). In all experiments, a within-subject factor assessed changes across days, trials, or blocks of trials using planned linear analyses. When using days, the average responding for each relevant period (see previous paragraph) in each group was used. Trials or blocks of trials was used for forward fear conditioning, fear and appetitive extinction, test, and backward conditioning in the last experiment. Experiment 4 employed a within-subject factor comparing freezing during light and light+tone.

All analyses were carried out using the PSY statistical software (School of Psychology, the University of New South Wales, Australia). The Type I error rate was controlled at alpha = 0.05 for each contrast tested. If interactions were detected, follow-up simple effects analyses were calculated to determine the source of the interactions. For each significant statistical comparison, we report measures of effect size partial eta-squared ($\eta^2$; $\eta^2$=0.01 is a small effect, $\eta^2$=0.06 a medium effect; and $\eta^2$=0.14 a large effect). The required number of rats per group was determined during the design stage of the study and was based on our prior studies on inhibitory learning (*Laurent et al., 2022*; *Lingawi et al., 2018*; *Laurent et al., 2018*; *Lingawi et al., 2017*).

## Acknowledgements

The research reported in this manuscript was supported by a Discovery Project Grant from the Australian Research Council (DP230101463) to NWL, VL, and MEB.

## Additional information

### Competing interests

Nathan Holmes: Reviewing editor, eLife. The other authors declare that no competing interests exist.

### Funding

| Funder | Grant reference number | Author |
| --- | --- | --- |
| Australian Research Council | DP230101463 | Nura W Lingawi Mark E Bouton Vincent Laurent |

The funders had no role in study design, data collection and interpretation, or the decision to submit the work for publication.

### Author contributions

Nura W Lingawi, Conceptualization, Data curation, Formal analysis, Funding acquisition, Investigation, Methodology, Writing – original draft, Writing – review and editing; Billy Chieng, Investigation, Methodology; R Fred Westbrook, Nathan Holmes, Conceptualization, Writing – review and editing; Mark E Bouton, Conceptualization, Funding acquisition, Writing – review and editing; Vincent Laurent, Conceptualization, Resources, Data curation, Formal analysis, Supervision, Funding acquisition, Investigation, Methodology, Writing – original draft, Project administration, Writing – review and editing

### Author ORCIDs

Nura W Lingawi  https://orcid.org/0000-0003-2455-0405
Nathan Holmes  https://orcid.org/0000-0002-0592-2026
Vincent Laurent  https://orcid.org/0000-0003-2333-8437

### Ethics

All the animals were handled according to approved Animal Care and Ethics Committee (ACEC) protocols of the University of New South Wales. The experimental protocols were approved by the UNSW ACEC (Permit Number: 21/28B). All surgery was performed under isoflurane anesthesia, and every effort was made to minimize suffering.

Reviewer #1 (Public review): https://doi.org/10.7554/eLife.108719.3.sa1
Reviewer #2 (Public review): https://doi.org/10.7554/eLife.108719.3.sa2
Reviewer #3 (Public review): https://doi.org/10.7554/eLife.108719.3.sa3
Author response https://doi.org/10.7554/eLife.108719.3.sa4

## Additional files

### Supplementary files
MDAR checklist

### Data availability
All data generated and/or analysed during this study are available via the Open Science Framework repository (https://osf.io/wr7gx/). The files contain the numerical data used to generate the figures.

The following dataset was generated:

| Author(s) | Year | Dataset title | Dataset URL | Database and Identifier |
|---|---|---|---|---|
| Laurent V | 2025 | Source data for eLife article "Backward Conditioning Reveals Flexibility in Infralimbic Cortex Inhibitory Memories" | https://osf.io/wr7gx/ | Open Science Framework, wr7gx |

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
