## [Editor Report · eLife Assessment]

This set of experiments provides **important** knowledge for how the infralimbic cortex is recruited to inhibit behavior after extinction training. The evidence supporting the conclusions is **convincing** with multiple sophisticated behavioral designs providing converging lines of evidence, though reviewers note possible alternative interpretations and limitations of small group sizes in some cases. This work will be of interest to those interested in cortical function, learning and memory, aversive behavior, and/or related psychiatric factors.

---

## [Referee Report · Reviewer #1 (Public review)]

The revised manuscript presents an interesting and technically competent set of experiments exploring the role of the infralimbic cortex (IL) in extinction learning. The inclusion of histological validation in the supplemental material improves the transparency and credibility of the results, and the overall presentation has been clarified. However, several key issues remain that limit the strength of the conclusions.

The behavioral effects reported are modest, as evident from the trial-by-trial data included in the supplemental figures. Although the authors interpret their findings as evidence that IL stimulation facilitates extinction only after prior inhibitory learning, this conclusion is not directly supported by their data. The experiments do not include a condition in which IL stimulation is delivered during extinction training alone, without prior inhibitory experience. Without this control, the claim that prior inhibitory memory is necessary for facilitation remains speculative.

The electrophysiological example provided shows that IL stimulation induces a sustained inhibition that outlasts the stimulation period. This prolonged suppression could potentially interfere with consolidation processes following tone presentation rather than facilitating them. The authors should consider and discuss this alternative interpretation in light of their behavioral data.

It is unfortunate that several animals had to be excluded after histological verification, but the resulting mismatch between groups remains a concern. Without a power analysis indicating the number of subjects required to achieve reliable effects, it is difficult to determine whether the modest behavioral differences reflect genuine biological variability or insufficient statistical power. Additional animals may be needed to properly address this imbalance.

Overall, while the manuscript is improved in clarity and methodological detail, the behavioral effects remain weak, and the mechanistic interpretation requires stronger experimental support and consideration of alternative explanations.

---

## [Referee Report · Reviewer #2 (Public review)]

Summary:

In this manuscript, the authors examine the mechanisms by which stimulation of the infralimbic cortex (IL) facilitates the retention and retrieval of inhibitory memories. Previous work has shown that optogenetic stimulation of the IL suppresses freezing during extinction but does not improve extinction recall when extinction memory is probed one day later. When stimulation occurs during a second extinction session (following a prior stimulation-free extinction session), freezing is suppressed during the second extinction as well as during the tone test the following day. The current study was designed to further explore the facilitatory role of the IL in inhibitory learning and memory recall. The authors conducted a series of experiments to determine whether recruitment of IL extends to other forms of inhibitory learning (e.g., backward conditioning) and to inhibitory learning involving appetitive conditioning. Further, they assessed whether their effects could be explained by stimulus familiarity. The results of their experiments show that backward conditioning, another form of inhibitory learning, also enabled IL stimulation to enhance fear extinction. This phenomenon was not specific to aversive learning as backward appetitive conditioning similarly allowed IL stimulation to facilitate extinction of aversive memories. Finally, the authors ruled out the possibility that IL facilitated extinction merely because of prior experience with the stimulus (e.g., reducing the novelty of the stimulus). These findings significantly advance our understanding of the contribution of IL to inhibitory learning. Namely, they show that the IL is recruited during various forms of inhibitory learning and its involvement is independent of the motivational value associated with the unconditioned stimulus.

Strengths to highlight:

(1) Transparency about the inclusion of both sexes and the representation of data from both sexes in figures.

(2) Very clear representation of groups and experimental design for each figure.

(3) The authors were very rigorous in determining the neurobehavioral basis for the effects of IL stimulation on extinction. They considered multiple interpretations and designed experiments to address these possible accounts of their data.

(4) The rationale for and the design of the experiments in this manuscript are clearly based on a wealth of knowledge about learning theory. The authors leveraged this expertise to narrow down how the IL encodes and retrieves inhibitory memories.

---

## [Referee Report · Reviewer #3 (Public review)]

Summary:

This is a really nice manuscript with different lines of evidence to show that the IL encodes inhibitory memories that can then be manipulated by optogenetic stimulation of these neurons during extinction. The behavioral designs are excellent, with converging evidence using extinction/re-extinction, backwards/forwards aversive conditioning, and backwards appetitive/forwards aversive conditioning. Additional factors, such as nonassociative effects of the CS or US, also are considered, and the authors evaluate the inhibitory properties of the CS with tests of conditioned inhibition. The authors have addressed the prior reviews. I still think it is unfortunate that the groups were not properly balanced in some of the figures (as noted by the authors, they were matched appropriately in real time, but some animals had to be dropped after histology, which caused some balancing issues). I think the overall pattern of results is compelling enough that more subjects do not need to be added, but it would still be nice to see more acknowledgement and statistical analyses of how these pre-existing differences may have impacted test performance.

Strengths:

The experimental designs are very rigorous with an unusual level of behavioral sophistication.

Weaknesses:

The various group differences in Figure 2 prior to any manipulation are still problematic. There was a reliable effect of subsequent group assignment in Figure 2 (p<0.05, described as "marginal" in multiple places). Then there are differences in extinction (nonsignificant at p=.07). The test difference between ReExt OFF/ON is identical to the difference at the end of extinction and the beginning of Forward 2, in terms of absolute size. I really don't think much can be made of the test result. The authors state in their response that this difference was not evident during the forward phase, but there clearly is a large ordinal difference on the first trial. I think it is appropriate to only focus on test differences when groups are appropriately matched, but when there are pre-existing differences (even when not statistically significant) then they really need to be incorporated into the statistical test somehow.

The same problem is evident in Figure 4B, but here the large differences in the Same groups are opposite to the test differences. It's hard to say how those large differences ultimately impacted the test results. I suppose it is good that the differences during Forward conditioning did not ultimately predict test differences, but this really should have been addressed with more subjects in these experiments. The authors explore the interactions appropriately but with n=6 in the various subgroups, it's not surprising that some of these effects were not detected statistically.

It is useful to see the trial-by-trial test data now presented in the supplement. I think the discussion does a good job of addressing the issues of retrieval, but the ideas of Estes about session cues that the authors bring up in their response haven't really held up over the years (e.g., Robbins, 1990, who explicitly tested this; other demonstrations of within-session spontaneous recovery), for what it's worth.

---

## [Author Response]

The following is the authors’ response to the current reviews.

**Public Reviews:**

**Reviewer #1 (Public review):**
The revised manuscript presents an interesting and technically competent set of experiments exploring the role of the infralimbic cortex (IL) in extinction learning. The inclusion of histological validation in the supplemental material improves the transparency and credibility of the results, and the overall presentation has been clarified. However, several key issues remain that limit the strength of the conclusions.

We thank the Reviewer for their positive assessment of our revised manuscript. We discussed the issues raised by the Reviewer below.

The behavioral effects reported are modest, as evident from the trial-by-trial data included in the supplemental figures. Although the authors interpret their findings as evidence that IL stimulation facilitates extinction only after prior inhibitory learning, this conclusion is not directly supported by their data. The experiments do not include a condition in which IL stimulation is delivered during extinction training alone, without prior inhibitory experience. Without this control, the claim that prior inhibitory memory is necessary for facilitation remains speculative.

The manuscript provides evidence across five experiments (Figures 2-6) that IL stimulation fails to facilitate extinction training in the absence of prior inhibitory experience. We therefore remain confident that the data support our conclusion: prior inhibitory learning enables IL stimulation to facilitate subsequent inhibitory learning.

The electrophysiological example provided shows that IL stimulation induces a sustained inhibition that outlasts the stimulation period. This prolonged suppression could potentially interfere with consolidation processes following tone presentation rather than facilitating them. The authors should consider and discuss this alternative interpretation in light of their behavioral data.

The possibility that IL stimulation exerted its effects by interfering with consolidation processes is inconsistent with the literature. Disrupting consolidation processes in the IL impairs extinction learning (1), even when animals have prior inhibitory learning experience (2). Yet our experiments found that IL stimulation failed to interfere with initial extinction learning but instead facilitated subsequent learning. Furthermore, the electrophysiological example demonstrates that the inhibitory effect is transient: the cell returned to firing properties similar to those observed pre-stimulation, making it unlikely that inhibition persists during the consolidation window.

It is unfortunate that several animals had to be excluded after histological verification, but the resulting mismatch between groups remains a concern. Without a power analysis indicating the number of subjects required to achieve reliable effects, it is difficult to determine whether the modest behavioral differences reflect genuine biological variability or insufficient statistical power. Additional animals may be needed to properly address this imbalance.

As noted in the revised manuscript, we are confident about the reliability of the findings reported. The manuscript provides evidence across five experiments that IL stimulation fails to facilitate brief extinction in the absence of prior inhibitory experience, replicating previous findings (3, 4). The manuscript also replicates these prior studies by demonstrating that experience with either fear or appetitive extinction enables IL stimulation to facilitate subsequent fear extinction. Furthermore, the present experiments replicate the facilitative effects of IL stimulation following fear or appetitive backward conditioning.

Overall, while the manuscript is improved in clarity and methodological detail, the behavioral effects remain weak, and the mechanistic interpretation requires stronger experimental support and consideration of alternative explanations.

We respectfully disagree with the assertion that the reported results are weak. The manuscript replicates all main findings internally or reproduces findings from previously published studies. While alternative explanations cannot be entirely excluded, we are not aware of any competing account that predicts the pattern of results reported here.

**Reviewer #2 (Public review):**
Summary:In this manuscript, the authors examine the mechanisms by which stimulation of the infralimbic cortex (IL) facilitates the retention and retrieval of inhibitory memories. Previous work has shown that optogenetic stimulation of the IL suppresses freezing during extinction but does not improve extinction recall when extinction memory is probed one day later. When stimulation occurs during a second extinction session (following a prior stimulation-free extinction session), freezing is suppressed during the second extinction as well as during the tone test the following day. The current study was designed to further explore the facilitatory role of the IL in inhibitory learning and memory recall. The authors conducted a series of experiments to determine whether recruitment of IL extends to other forms of inhibitory learning (e.g., backward conditioning) and to inhibitory learning involving appetitive conditioning. Further, they assessed whether their effects could be explained by stimulus familiarity. The results of their experiments show that backward conditioning, another form of inhibitory learning, also enabled IL stimulation to enhance fear extinction. This phenomenon was not specific to aversive learning as backward appetitive conditioning similarly allowed IL stimulation to facilitate extinction of aversive memories. Finally, the authors ruled out the possibility that IL facilitated extinction merely because of prior experience with the stimulus (e.g., reducing the novelty of the stimulus). These findings significantly advance our understanding of the contribution of IL to inhibitory learning. Namely, they show that the IL is recruited during various forms of inhibitory learning and its involvement is independent of the motivational value associated with the unconditioned stimulus.

We thank the Reviewer for their positive assessment.

Strengths to highlight:(1) Transparency about the inclusion of both sexes and the representation of data from both sexes in figures

We thank the Reviewer for their positive assessment.

(2) Very clear representation of groups and experimental design for each figure

We thank the Reviewer for their positive assessment.

(3) The authors were very rigorous in determining the neurobehavioral basis for the effects of IL stimulation on extinction. They considered multiple interpretations and designed experiments to address these possible accounts of their data.

We thank the Reviewer for their positive assessment.

(4) The rationale for and the design of the experiments in this manuscript are clearly based on a wealth of knowledge about learning theory. The authors leveraged this expertise to narrow down how the IL encodes and retrieves inhibitory memories.

We thank the Reviewer for their positive assessment.

**Reviewer #3 (Public review):**
Summary:This is a really nice manuscript with different lines of evidence to show that the IL encodes inhibitory memories that can then be manipulated by optogenetic stimulation of these neurons during extinction. The behavioral designs are excellent, with converging evidence using extinction/re-extinction, backwards/forwards aversive conditioning, and backwards appetitive/forwards aversive conditioning. Additional factors, such as nonassociative effects of the CS or US, also are considered, and the authors evaluate the inhibitory properties of the CS with tests of conditioned inhibition. The authors have addressed the prior reviews. I still think it is unfortunate that the groups were not properly balanced in some of the figures (as noted by the authors, they were matched appropriately in real time, but some animals had to be dropped after histology, which caused some balancing issues). I think the overall pattern of results is compelling enough that more subjects do not need to be added, but it would still be nice to see more acknowledgement and statistical analyses of how these pre-existing differences may have impacted test performance.

We thank the Reviewer for their positive assessment of our revised manuscript. We discussed the comments regarding group balancing below.

Strengths:The experimental designs are very rigorous with an unusual level of behavioral sophistication.

We thank the Reviewer for their positive assessment

Weaknesses:The various group differences in Figure 2 prior to any manipulation are still problematic. There was a reliable effect of subsequent group assignment in Figure 2 (p<0.05, described as "marginal" in multiple places). Then there are differences in extinction (nonsignificant at p=.07). The test difference between ReExt OFF/ON is identical to the difference at the end of extinction and the beginning of Forward 2, in terms of absolute size. I really don't think much can be made of the test result. The authors state in their response that this difference was not evident during the forward phase, but there clearly is a large ordinal difference on the first trial. I think it is appropriate to only focus on test differences when groups are appropriately matched, but when there are pre-existing differences (even when not statistically significant) then they really need to be incorporated into the statistical test somehow.

We carefully considered the Reviewer's suggestion, but it is not possible to adjust the statistical analyses at test because these analyses do not directly compare the two ReExt groups. Any scaling of performance would require including the two Ext groups, which is not feasible since these groups did not receive initial extinction. Moreover, the analyses provide no conclusive evidence of pre-existing differences between the two ReExt groups: the difference was not significant during initial extinction and was absent during the Forward 2 stage. We acknowledge that closer performance between the two ReExt groups during initial extinction would have been preferable. However, we remain confident in the results obtained because they replicate previous experiments in which the two ReExt groups displayed identical performance during initial extinction.

The same problem is evident in Figure 4B, but here the large differences in the Same groups are opposite to the test differences. It's hard to say how those large differences ultimately impacted the test results. I suppose it is good that the differences during Forward conditioning did not ultimately predict test differences, but this really should have been addressed with more subjects in these experiments. The authors explore the interactions appropriately but with n=6 in the various subgroups, it's not surprising that some of these effects were not detected statistically.

As the Reviewer noted, the unexpected differences in Figure 4B are opposite in direction to the test differences. Importantly, Figure 4B replicates the main findings from Figure 3, which did not show these unexpected differences.

It is useful to see the trial-by-trial test data now presented in the supplement. I think the discussion does a good job of addressing the issues of retrieval, but the ideas of Estes about session cues that the authors bring up in their response haven't really held up over the years (e.g., Robbins, 1990, who explicitly tested this; other demonstrations of within-session spontaneous recovery), for what it's worth.

We thank the Reviewer for bringing our attention to Robbins’ work on session cues. We understand that the issue of retrieval is important but as we noted before, our manuscript and its conclusions do not claim to differentiate retrieval from additional learning.

References

(1) K. E. Nett, R. T. LaLumiere, Infralimbic cortex functioning across motivated behaviors: Can the differences be reconciled Neurosci Biobehav Rev 131, 704–721 (2021).

(2) V. Laurent, R. F. Westbrook, Inactivation of the infralimbic but not the prelimbic cortex impairs consolidation and retrieval of fear extinction Learn Mem 16, 520–529 (2009).

(3) N. W. Lingawi, R. F. Westbrook, V. Laurent, Extinction and Latent Inhibition Involve a Similar Form of Inhibitory Learning that is Stored in and Retrieved from the Infralimbic Cortex Cereb Cortex 27, 5547–5556 (2017).

(4) N. W. Lingawi, N. M. Holmes, R. F. Westbrook, V. Laurent, The infralimbic cortex encodes inhibition irrespective of motivational significance Neurobiol Learn Mem 150, 64–74 (2018).

The following is the authors’ response to the original reviews.

**Public Reviews:**

**Reviewer #1 (Public review):**
Summary:The manuscript reports a series of experiments designed to test whether optogenetic activation of infralimbic (IL) neurons facilitates extinction retrieval and whether this depends on animals' prior experience. In Experiment 1, rats underwent fear conditioning followed by either one or two extinction sessions, with IL stimulation given during the second extinction; stimulation facilitated extinction retrieval only in rats with prior extinction experience. Experiments 2 and 3 examined whether backward conditioning (CS presented after the US) could establish inhibitory properties that allowed IL stimulation to enhance extinction, and whether this effect was specific to the same stimulus or generalized to different stimuli. Experiments 5 - 7 extended this approach to appetitive learning: rats received backward or forward appetitive conditioning followed by extinction, and then fear conditioning, to determine whether IL stimulation could enhance extinction in contexts beyond aversive learning and across conditioning sequences. Across studies, the key claim is that IL activation facilitates extinction retrieval only when animals possess a prior inhibitory memory, and that this effect generalizes across aversive and appetitive paradigms.Strengths:(1) The design attempts to dissect the role of IL activity as a function of prior learning, which is conceptually valuable.

We thank the Reviewer for their positive assessment.

(2) The experimental design of probing different inhibitory learning approaches to probe how IL activation facilitates extinction learning was creative and innovative.

We thank the Reviewer for their positive assessment.

Weaknesses:(1) Non-specific manipulation.ChR2 was expressed in IL without distinction between glutamatergic and GABAergic populations. Without knowing the relative contribution of these cell types or the percentage of neurons affected, the circuit-level interpretation of the results is unclear.

ChR2 was intentionally expressed in the infralimbic cortex (IL) without distinction between local neuronal populations for two reasons. First, the primary aim of this was to uncover some of the features characterizing the encoding of inhibitory memories in the IL, and this encoding likely engages interactions among various neuronal populations within the IL. Second, the hypotheses tested in the manuscript derived from findings that indiscriminately stimulated the IL using the GABA_A_ receptor antagonist picrotoxin, which is best mimicked by the approach taken. We agree that it is also important to determine the respective contributions of distinct IL neuronal populations to inhibitory encoding; however, the global approach implemented in the present experiments represents a necessary initial step. These matters have been incorporated in the Discussion of the revised manuscript.

(2) Extinction retrieval test conflates processesThe retrieval test included 8 tones. Averaging across this many tone presentations conflate extinction retrieval/expression (early tones) with further extinction learning (later tones). A more appropriate analysis would focus on the first 2-4 tones to capture retrieval only. As currently presented, the data do not isolate extinction retrieval.

It is unclear when retrieval of what has been learned across extinction ceases and additional extinction learning occurs. In fact, it is only the first stimulus presentation that unequivocally permits a distinction between retrieval and additional extinction learning, as the conditions for this additional learning have not been fulfilled at that presentation. However, confining evidence for retrieval to the first stimulus presentation introduces concerns that other factors could influence performance. For instance, processing of the stimulus present at the start of the session may differ from that present at the end of the previous session, thereby affecting what is retrieved. Such differences between the stimuli present at the start and end of an extinction session have been long recognized as a potential explanation for spontaneous recovery (Estes, 1955). More importantly, whether the test data presented confound retrieval and additional extinction learning or not, the interpretation remains the same with respect to the effects of a prior history of inhibitory learning on enabling the facilitative effects of IL stimulation. Finally, it is unclear how these facilitative effects could occur in the absence of the subjects retrieving the extinction memory formed under the stimulation. Nevertheless, the revised manuscript now provides the trial-by-trial performance (see Supplemental Figure 3) during the post-extinction retrieval tests and addresses this issue in the Discussion.

(3) Under-sampling and poor group matching.Sample sizes appear small, which may explain why groups are not well matched in several figures (e.g., 2b, 3b, 6b, 6c) and why there are several instances of unexpected interactions (protocol, virus, and period). This baseline mismatch raises concerns about the reliability of group differences.

Efforts were made to match group performance upon completion of each training stage and before IL stimulation. Unfortunately, these efforts were not completely successful due to exclusions following post-mortem analyses. This has been made explicit in the revised manuscript (Materials and Methods, Subjects section). However, we acknowledge that the unexpected interactions deserve further discussion, and this has been incorporated into the revised manuscript (see also comment from Reviewer 2). Although we cannot exclude the possibility that sample sizes may have contributed to some of these interactions, we remain confident about the reliability of the main findings reported, especially given their replication across the various protocols. Overall, the manuscript provides evidence that IL stimulation does not facilitate brief extinction in the absence of prior inhibitory experience in five different experiments, replicating previous findings (Lingawi et al., 2018; Lingawi et al., 2017). It also replicates these previous findings by showing that prior experience with either fear or appetitive extinction enables IL stimulation to facilitate subsequent fear extinction. Furthermore, the facilitative effects of such stimulation following fear or appetitive backward conditioning are replicated in the present manuscript. This is discussed in the Discussion of the revised manuscript.

(4) Incomplete presentation of conditioning dataFigure 3 only shows a single conditioning session despite five days of training. Without the full dataset, it is difficult to evaluate learning dynamics or whether groups were equivalent before testing.

We apologize, as we incorrectly labeled the X axis for the backward conditioning data in Figures 3B, 4B, 4D and 5B. It should have indicated “Days” instead of “Trials”. This error has been corrected in the revised manuscript (see also second comment from Reviewer 2).

(5) Interpretation stronger than evidence.The authors conclude that IL activation facilitates extinction retrieval only when an inhibitory memory has been formed. However, given the caveats above, the data are insufficient to support such a strong mechanistic claim. The results could reflect nonspecific facilitation or disruption of behavior by broad prefrontal activation. Moreover, there is compelling evidence that optogenetic activation of IL during fear extinction does facilitate subsequent extinction retrieval without prior extinction training (DoMonte et al 2015, Chen et al 2021), which the authors do not directly test in this study.

As noted above, the interpretations of the main findings stand whether the test data confounds retrieval with additional extinction learning or not. The revised manuscript also clarifies the plotting of the data for the backward conditioning stages. We do agree that further discussion of the unexpected interactions is necessary, and this has been incorporated into the revised manuscript. However, the various replications of the core findings provide strong evidence for their reliability and the interpretations advanced in the original manuscript. The proposal that the results reflect non-specific facilitation or disruption of behavior seems highly unlikely. Indeed, the present experiments and previous findings (Lingawi et al., 2018; Lingawi et al., 2017) provide multiple demonstrations that IL stimulation fails to produce any facilitation in the absence of prior inhibitory experience with the target stimulus. Although these demonstrations appear inconsistent with previous studies (Do-Monte et al., 2015; Chen et al., 2021), this inconsistency is likely explained by the fact that these studies manipulated activity in specific IL neuronal populations. Previous work has already revealed differences between manipulations targeting discrete IL neuronal populations as opposed to general IL activity (Kim et al., 2016). Importantly, as previously noted, the present manuscript aimed to generally explore inhibitory encoding in the IL that is likely to engage several neuronal populations within the IL. Adequate statements on these matters have been included in the Discussion of the revised manuscript.

**Reviewer #2 (Public review):**
Summary:In this manuscript, the authors examine the mechanisms by which stimulation of the infralimbic cortex (IL) facilitates the retention and retrieval of inhibitory memories. Previous work has shown that optogenetic stimulation of the IL suppresses freezing during extinction but does not improve extinction recall when extinction memory is probed one day later. When stimulation occurs during a second extinction session (following a prior stimulation-free extinction session), freezing is suppressed during the second extinction as well as during the tone test the following day. The current study was designed to further explore the facilitatory role of the IL in inhibitory learning and memory recall. The authors conducted a series of experiments to determine whether recruitment of IL extends to other forms of inhibitory learning (e.g., backward conditioning) and to inhibitory learning involving appetitive conditioning. Further, they assessed whether their effects could be explained by stimulus familiarity. The results of their experiments show that backward conditioning, another form of inhibitory learning, also enabled IL stimulation to enhance fear extinction. This phenomenon was not specific to aversive learning, as backward appetitive conditioning similarly allowed IL stimulation to facilitate extinction of aversive memories. Finally, the authors ruled out the possibility that IL facilitated extinction merely because of prior experience with the stimulus (e.g., reducing the novelty of the stimulus). These findings significantly advance our understanding of the contribution of IL to inhibitory learning. Namely, they show that the IL is recruited during various forms of inhibitory learning, and its involvement is independent of the motivational value associated with the unconditioned stimulus.Strengths:(1) Transparency about the inclusion of both sexes and the representation of data from both sexes in figures.

We thank the Reviewer for their positive assessment.

(2) Very clear representation of groups and experimental design for each figure.

We thank the Reviewer for their positive assessment.

(3) The authors were very rigorous in determining the neurobehavioral basis for the effects of IL stimulation on extinction. They considered multiple interpretations and designed experiments to address these possible accounts of their data.

We thank the Reviewer for their positive assessment.

(4) The rationale for and the design of the experiments in this manuscript are clearly based on a wealth of knowledge about learning theory. The authors leveraged this expertise to narrow down how the IL encodes and retrieves inhibitory memories.

We thank the Reviewer for their positive assessment.

Weaknesses:(1) In Experiment 1, although not statistically significant, it does appear as though the stimulation groups (OFF and ON) differ during Extinction 1. It seems like this may be due to a difference between these groups after the first forward conditioning. Could the authors have prevented this potential group difference in Extinction 1 by re-balancing group assignment after the first forward conditioning session to minimize the differences in fear acquisition (the authors do report a marginally significant effect between the groups that would undergo one vs. two extinction sessions in their freezing during the first conditioning session)?

Efforts were made daily to match group performance across the training stages, but these efforts were ultimately hampered by the necessary exclusions following postmortem analyses. This has been made explicit in the revised manuscript (Materials and Methods, Subjects section). Regarding freezing during Extinction 1, as noted by the Reviewer, the difference, which was not statistically significant, was absent across trials during the subsequent forward fear conditioning stage. Likewise, the protocol difference observed during the initial forward fear conditioning was absent in subsequent stages. We are therefore confident that these initial differences (significant or not) did not impact the main findings at test. Importantly, these findings replicate previous work using identical protocols in which no differences were present during the training stages. These considerations have been addressed in the revised manuscript (see Results for Experiment 1).

(2) Across all experiments (except for Experiment 1), the authors state that freezing during the initial conditioning increased across "days". The figures that correspond to this text, however, show that freezing changes across trials. In the methods, the authors report that backward conditioning occurred over 5 days. It would be helpful to understand how these data were analyzed and collated to create the final figures. Was the freezing averaged across the five days for each trial for analyses and figures?

We apologize, as noted above, for having incorrectly labeled the X axis across the backward conditioning data sets in Figures 3B, 4B, 4D and 5B. It should have indicated “Days” instead of “Trials”. The data shown in these Figures use the average of all trials on a given day. This has been clarified in the methods section of the revised manuscript (Statistical Analyses section). The labeling errors on the Figures have been corrected.

(3) In Experiment 3, the authors report a significant Protocol X Virus interaction. It would be useful if the authors could conduct post-hoc analyses to determine the source of this interaction. Inspection of Figure 4B suggests that freezing during the two different variants of backward conditioning differs between the virus groups. Did the authors expect to see a difference in backward conditioning depending on the stimulus used in the conditioning procedure (light vs. tone)? The authors don't really address this confounding interaction, but I do think a discussion is warranted.

We agree with the Reviewer that further discussion of the Protocol x Virus interaction that emerged during the backward conditioning and forward conditioning stages of Experiment 3 is warranted. This discussion has been provided in the revised manuscript (see Results section). Briefly, during both stages, follow-up analyses did not reveal any differences (main effects or interactions) between the two groups trained with the light stimulus (Diff-EYFP and Diff-ChR2). By contrast, the ChR2 group trained with the tone (Back-ChR2) froze more overall than the EYFP group (Back-EYFP), but there were no other significant differences between the two groups. Based on these analyses, the Protocol x Virus interaction appears to be driven by greater freezing in the ChR2 group trained with the tone rather than a difference in the backward conditioning performance based on stimulus identity. Consistent with this, the statistical analyses did not reveal a main effect of Protocol during either the backward conditioning stage or the stimulus trials during the forward conditioning stage. Nevertheless, during this latter stage, a main effect of Protocol emerged during baseline performance, but once again, this seems to be driven by the Back-ChR2 group. Critically, it is unclear how greater stimulus freezing in the Back-ChR2 group during forward conditioning would lead to lower freezing during the post-extinction retrieval test.

We note that an unexpected Protocol x Period interaction was found during appetitive backward conditioning in Experiment 5. For consistency, we conducted additional analyses to determine the source of this interaction (see Results section). As previously noted, performance during appetitive backward conditioning is noisy and cannot be taken as a failure to generate inhibitory learning. It is therefore unlikely that this interaction implied a difference in such learning.

(4) In this same experiment, the authors state that freezing decreased during extinction; however, freezing in the Diff-EYFP group at the start of extinction (first bin of trials) doesn't look appreciably different than their freezing at the end of the session. Did this group actually extinguish their fear? Freezing on the tone test day also does not look too different from freezing during the last block of extinction trials.

We confirm that overall, there was a significant decline in freezing across the extinction session shown in Figure 4B. The Reviewer is correct to point out that this decline was modest (if not negligible) in the Diff-EYFP group, which was receiving its first inhibitory training with the target tone stimulus. It is worth noting that across all experiments, most groups that did not receive infralimbic stimulation displayed a modest decline in freezing during the extinction session since it was relatively brief, involving only 6 or 8 tone alone presentations. This was intentional, as we aimed for the brief extinction session to generate minimal inhibitory learning and thereby to detect any facilitatory effect of infralimbic stimulation. This has been clarified and explained in the revised version of the manuscript (see Results section, description of Experiment 1).

(5) The Discussion explored the outcomes of the experiments in detail, but it would be useful for the authors to discuss the implications of their findings for our understanding of circuits in which the IL is embedded that are involved in inhibitory learning and memory. It would also be useful for the authors to acknowledge in the Discussion that although they did not have the statistical power to detect sex differences, future work is needed to explore whether IL functions similarly in both sexes.

In line with the Reviewer’s suggestion (see also Reviewer 3), the Discussion section has been substantially altered in the revised manuscript. Among other things, it does mention that future studies will need to examine the role of additional brain regions in the effects reported and it acknowledges the need to further explore sex differences and IL functions.

**Reviewer #3 (Public review):**
Summary:This is a really nice manuscript with different lines of evidence to show that the IL encodes inhibitory memories that can then be manipulated by optogenetic stimulation of these neurons during extinction. The behavioral designs are excellent, with converging evidence using extinction/re-extinction, backwards/forwards aversive conditioning, and backwards appetitive/forwards aversive conditioning. Additional factors, such as nonassociative effects of the CS or US, are also considered, and the authors evaluate the inhibitory properties of the CS with tests of conditioned inhibition.Strengths:The experimental designs are very rigorous with an unusual level of behavioral sophistication.

We thank the Reviewer for their positive assessment

Weaknesses:(1) More justification for parametric choices (number of days of backwards vs forwards conditioning) could be provided.

All experimental parameters were based on previously published experiments showing the capacity of the backward conditioning protocols to generate inhibitory learning and the forward conditioning protocols to produce excitatory learning. Although this was mentioned in the methods section, we acknowledge that further explanation was required to justify the need for multiple days of backward training. This has been provided in the revised manuscript (see Results section and description of the backward parameters).

(2) The current discussion could be condensed and could focus on broader implications for the literature.

The discussion has been severely condensed and broader implications have been discussed with respect to the existing literature looking at the neural circuitry underlying inhibitory learning.

**Recommendations for the authors:**

**Reviewer #1 (Recommendations for the authors):**
(1) Re-analyze extinction retrieval, focusing only on the first 2-4 tones to capture extinction expression.

This recommendation corresponds to the second public comment made by the Reviewer, and we have replied to this comment.

(2) Directly test whether activation of IL during fear extinction is insufficient to facilitate extinction retrieval without prior extinction training.

The manuscript provides five separate demonstrations that the optogenetic approach to stimulate IL activity did not facilitate the initial brief extinction session. This reproduces what had been found with indiscriminate pharmacological stimulation in our previous research (Lingawi et al., 2018; Lingawi et al., 2017). We appreciate that other work that stimulated specific IL neuronal populations has observed facilitation of extinction but, the present manuscript focuses on the role of all IL neuronal populations in encoding inhibitory memories. The Reviewer’s request would imply contrasting the role of various neuronal populations, which is beyond the scope of this manuscript. Nevertheless, we have modified our discussion to indicate that future research should establish which IL neuronal population(s) contribute to the effects reported here.

(3) Show the percentage of neurons that exhibit excitatory or inhibitory responses in IL after non-specific optogenetic activation to better understand how this manipulation is affecting IL circuitry.

All electrophysiological recordings (n = 10 cells) are presented in Figure 1C. ChR2 excitation was substantial and overwhelming. Based on the physiological and morphological characteristics of the recorded cells, one was non-pyramidal and was excited by LED light delivery. The remaining 9 cells were pyramidal. One did not respond to LED delivery, but we cannot exclude the possibility that this was due to a lack of ChR2 expression in the somatic compartment. Another cell showed a mild reduction in activity following LED stimulation, while the remaining 7 cells displayed clear excitation upon LED stimulation. We have modified our manuscript to reflect these observations. We did not include percentages since only 10 recordings are shown.

(4) Present data from all five conditioning sessions, not just one, to allow evaluation of learning history.

This recommendation corresponds to the fourth public comment made by the Reviewer, and we have replied to this comment.

(5) Address the issue of small and poorly matched groups, particularly in Figures 2b, 3b, 6b, and 6c.

This recommendation corresponds to the third public comment made by the Reviewer, and we have replied to this comment.

(6) Temper the conclusions to reflect the limitations of sampling, group matching, and the lack of specificity in the manipulation.

We have modified our Discussion to address potential issues related to sampling and group matching. However, we are unsure how the lack of specificity of the IL stimulation has any impact on the interpretations made, since no statement is made about neuronal specificity. That said, as noted above, “we have modified our discussion to indicate that future research should establish which IL neuronal population(s) contribute to the effects reported here”.

**Reviewer #2 (Recommendations for the authors):**
Nothing additional to include beyond what is written for public view.
**Reviewer #3 (Recommendations for the authors):**
This is a really nice manuscript with different lines of evidence to show that the IL encodes inhibitory memories that can then be manipulated by optogenetic stimulation of these neurons during extinction. The behavioral designs are excellent, with converging evidence using extinction/re-extinction, backwards/forwards aversive conditioning, and backwards appetitive/forwards aversive conditioning. Additional factors, such as nonassociative effects of the CS or US, are also considered, and the authors evaluate the inhibitory properties of the CS with tests of conditioned inhibition. I only have a couple of comments that the authors may want to consider.

We thank the Reviewer for their positive assessment.

First, in Figure 2, it is unfortunate that there is a general effect of the LED assignment before the LED experience (p=.07 during that first extinction session). This is in the same direction as the difference during the test, so it is not clear that the test difference really reflects differences due to Extinction 2 treatment or to preexisting differences based on group assignments.

The Reviewer’s comment is identical to the first public comment of Reviewer 2, which has been addressed.

Second, it is notable that the backwards fear conditioning phase was conducted over 5 days, but the forward conditioning phase was conducted over one day. The rationale for these differences should be presented. There is an old idea going back to Konorski that backwards conditioning may lead to excitation initially, and it is only after more extensive trials that inhibitory conditioning occurs (a finding supported by Heth, 1976). Some discussion of the potential biphasic nature of backwards conditioning would be useful, especially for people who want to run this type of experiment but with only a single session of backwards conditioning.

In line with the Reviewer’s suggestion, the revised manuscript (see results section) provide an explanation for conducting backward conditioning across multiple days.

Third, as written, each paragraph of the discussion is mostly a recapitulation of the findings from each experiment. This could be condensed significantly, and it would be nice to see more integration with the current literature and how these results challenge or suggest nuance in current thinking about IL function.

We have significantly condensed the recapitulation of our findings in the Discussion of the revised manuscript. The Discussion now dedicates space to address comments from the other Reviewers and integrate the present findings with the current literature.

References

Chen, Y.-H., Wu, J.-L., Hu, N.-Y., Zhuang, J.-P., Li, W.-P., Zhang, S.-R., Li, X.-W., Yang, J.-M., & Gao, T.-M. (2021). Distinct projections from the infralimbic cortex exert opposing effects in modulating anxiety and fear. J Clin Invest, 131(14), e145692. https://doi.org/10.1172/JCI145692

Do-Monte, F. H., Manzano-Nieves, G., Quiñones-Laracuente, K., Ramos-Medina, L., & Quirk, G. J. (2015). Revisiting the role of infralimbic cortex in fear extinction with optogenetics. J Neurosci, 35(8), 3607-3615. https://doi.org/10.1523/JNEUROSCI.3137-14.2015

Estes, W. K. (1955). Statistical theory of spontaneous recovery and regression. Psychol Rev, 62(3), 145-154. https://doi.org/10.1037/h0048509

Kim, H.-S., Cho, H.-Y., Augustine, G. J., & Han, J.-H. (2016). Selective Control of Fear Expression by Optogenetic Manipulation of Infralimbic Cortex after Extinction. Neuropsychopharmacology, 41(5), 1261-1273. https://doi.org/10.1038/npp.2015.276

Lingawi, N. W., Holmes, N. M., Westbrook, R. F., & Laurent, V. (2018). The infralimbic cortex encodes inhibition irrespective of motivational significance. Neurobiol Learn Mem, 150, 64-74. https://doi.org/10.1016/j.nlm.2018.03.001

Lingawi, N. W., Westbrook, R. F., & Laurent, V. (2017). Extinction and Latent Inhibition Involve a Similar Form of Inhibitory Learning that is Stored in and Retrieved from the Infralimbic Cortex. Cereb Cortex, 27(12), 5547-5556.

https://doi.org/10.1093/cercor/bhw322.